# Phagocytic glia are obligatory intermediates in transmission of mutant huntingtin aggregates across neuronal synapses

Kirby M Donnelly[1], Olivia R DeLorenzo[2], Aprem DA Zaya[1], Gabrielle E Pisano[1], Wint M Thu[1], Liqun Luo[3,4], Ron R Kopito[3], Margaret M Panning Pearce[1,2]*

[1]Department of Biological Sciences, University of the Sciences, Philadelphia, United States; [2]Program in Neuroscience, University of the Sciences, Philadelphia, United States; [3]Department of Biology, Stanford University, Stanford, United States; [4]Howard Hughes Medical Institute, Stanford University, Stanford, United States

**Abstract** Emerging evidence supports the hypothesis that pathogenic protein aggregates associated with neurodegenerative diseases spread from cell to cell through the brain in a manner akin to infectious prions. Here, we show that mutant huntingtin (mHtt) aggregates associated with Huntington disease transfer anterogradely from presynaptic to postsynaptic neurons in the adult *Drosophila* olfactory system. Trans-synaptic transmission of mHtt aggregates is inversely correlated with neuronal activity and blocked by inhibiting caspases in presynaptic neurons, implicating synaptic dysfunction and cell death in aggregate spreading. Remarkably, mHtt aggregate transmission across synapses requires the glial scavenger receptor Draper and involves a transient visit to the glial cytoplasm, indicating that phagocytic glia act as obligatory intermediates in aggregate spreading between synaptically-connected neurons. These findings expand our understanding of phagocytic glia as double-edged players in neurodegeneration—by clearing neurotoxic protein aggregates, but also providing an opportunity for prion-like seeds to evade phagolysosomal degradation and propagate further in the brain.

*For correspondence:
m.pearce@usciences.edu

Competing interests: The authors declare that no competing interests exist.

## Introduction

Neurodegenerative diseases have emerged as one of the greatest healthcare challenges in our aging society, and thus a better understanding of the underlying pathological mechanisms is critical for development of more effective treatments or cures for these fatal disorders. A common molecular feature of many neurodegenerative diseases [e.g., Alzheimer disease (AD), frontotemporal dementias (FTD), Parkinson disease (PD), amyotrophic lateral sclerosis (ALS), and Huntington disease (HD)] is the misfolding of certain proteins, driving their accumulation into insoluble, amyloid aggregates (*Knowles et al., 2014*). Appearance of proteinaceous deposits in patient brains correlates closely with neuronal loss and clinical progression, and strategies to lower production or enhance clearance of pathological proteins in the degenerating brain have shown therapeutic promise in animal models and clinical trials (*Boland et al., 2018*; *Li et al., 2019*; *Tabrizi et al., 2019*).

Post-mortem histopathological analyses (*Braak and Braak, 1991*; *Braak et al., 2003*; *Brettschneider et al., 2013*) and in vivo imaging studies (*Deng et al., 2004*; *Poudel et al., 2019*) indicate that proteopathic lesions associated with neurodegeneration appear in highly-reproducible and disease-specific spatiotemporal patterns through the brain. Interestingly, these patterns largely follow neuroanatomical tracts (*Ahmed et al., 2014*; *Mezias et al., 2017*), suggesting a central role for synaptic connectivity in pathological aggregate spreading. Accumulating evidence supports the

idea that intracellular aggregates formed by tau, α-synuclein, TDP-43, SOD1, and mutant huntingtin (mHtt) transfer from cell to cell and self-replicate by recruiting natively-folded versions of the same protein, analogous to how infectious prion protein (PrP$^{Sc}$) templates the conformational change of soluble PrP$^{C}$ in prion diseases (*Vaquer-Alicea and Diamond, 2019*). Numerous studies have pointed to roles for endocytosis and exocytosis (*Asai et al., 2015*; *Babcock and Ganetzky, 2015*; *Chen et al., 2019*; *Holmes et al., 2013*; *Lee et al., 2010*; *Zeineddine et al., 2015*), membrane permeabilization (*Chen et al., 2019*; *Falcon et al., 2018*; *Flavin et al., 2017*; *Zeineddine et al., 2015*), tunneling nanotubes (*Costanzo et al., 2013*; *Sharma and Subramaniam, 2019*), and neuronal activity (*Wu et al., 2016*) in entry and/or exit of pathogenic protein assemblies from cells, but the exact mechanisms by which amyloid aggregates or on-pathway intermediates cross one or more biological membranes in the highly complex central nervous system (CNS) remain an enigma.

Glia are resident immune cells of the CNS and constantly survey the brain to maintain homeostasis and respond rapidly to tissue damage or trauma. Reactive astrocytes and microglia provide a first line of defense in neurodegeneration by infiltrating sites of neuronal injury, upregulating immune-responsive genes, and phagocytosing dying neurons and other debris, including protein aggregates (*Asai et al., 2015*; *Grathwohl et al., 2009*; *Wyss-Coray et al., 2003*). Prolonged activation of these glial responses results in chronic inflammation, exacerbating synaptic dysfunction and neuronal loss (*Hammond et al., 2018*). We and others have previously shown that Draper, a *Drosophila* scavenger receptor that recognizes and phagocytoses cellular debris (*Freeman, 2015*), regulates the load of mHtt (*Pearce et al., 2015*) and Aβ$_{1-42}$ (*Ray et al., 2017*) aggregate pathology in the fly CNS. Remarkably, we also found that a portion of phagocytosed neuronal mHtt aggregates gain entry into the glial cytoplasm and once there, nucleate the aggregation of normally-soluble wild-type Htt (wtHtt) proteins, suggesting that glial phagocytosis provides a path for spreading of prion-like aggregates in intact brains. Consistent with these findings, microglial ablation suppresses pathological tau transmission between synaptically-connected regions of the mouse brain (*Asai et al., 2015*), and PrP$^{Sc}$ transfers from infected astrocytes to co-cultured neurons (*Victoria et al., 2016*). Thus, phagocytic glia may play double-edged roles in neurodegeneration, with normally neuroprotective clearance mechanisms also driving dissemination of prion-like aggregates through the brain.

A plethora of studies from the last decade have strengthened the prion-like hypothesis for neurodegenerative diseases, but we still lack a clear understanding of how pathogenic protein aggregates spread between cells in an intact CNS. In this study, we adapted our previously-described *Drosophila* HD model to investigate roles for synaptic connectivity and phagocytic glia in prion-like mHtt aggregate transmission in adult fly brains. HD is an autosomal dominant disorder caused by expansion of a CAG repeat region in exon 1 of the Htt gene, resulting in production of highly aggregation-prone mHtt proteins containing abnormally expanded polyglutamine (polyQ$\geq$37) tracts (*Bates et al., 2015*; *MacDonald et al., 1993*). By contrast, wtHtt proteins containing polyQ$\leq$36 tracts only aggregate upon nucleation by pre-formed Htt aggregate 'seeds' (*Chen et al., 2001*; *Preisinger et al., 1999*). A growing body of evidence from cell culture (*Chen et al., 2001*; *Costanzo et al., 2013*; *Holmes et al., 2013*; *Ren et al., 2009*; *Sharma and Subramaniam, 2019*; *Trevino et al., 2012*) and in vivo (*Ast et al., 2018*; *Babcock and Ganetzky, 2015*; *Jeon et al., 2016*; *Masnata et al., 2019*; *Pearce et al., 2015*; *Pecho-Vrieseling et al., 2014*) models of HD supports the idea that pathogenic mHtt aggregates have prion-like properties—they transfer from cell to cell and self-replicate by nucleating the aggregation of soluble wtHtt proteins. Here, we report that mHtt aggregates formed in presynaptic olfactory receptor neuron (ORN) axons effect prion-like conversion of wtHtt proteins expressed in the cytoplasm of postsynaptic partner projection neurons (PNs) in the adult fly olfactory system. Remarkably, transfer of mHtt aggregates from presynaptic ORNs to postsynaptic PNs was abolished in Draper-deficient animals and required passage of the prion-like aggregate seeds through the cytoplasm of phagocytic glial cells. Together, these findings support the conclusion that phagocytic glia are obligatory intermediates in prion-like transmission of mHtt aggregates between synaptically-connected neurons in vivo, providing new insight into key roles for glia in HD pathogenesis.

## Results

### Prion-like transfer of mHtt aggregates between synaptically-connected neurons in the adult fly olfactory system

Aggregates formed by N-terminal fragments of mHtt generated by aberrant splicing (e.g., exon 1; 'Htt$_{ex1}$') (*Sathasivam et al., 2013*) or caspase cleavage (e.g., exon 1–12; 'Htt$_{ex1-12}$') (*Graham et al., 2006*; *Figure 1A*) accumulate in HD patient brains, are highly cytotoxic, and spread between cells in culture and in vivo (*Babcock and Ganetzky, 2015*; *Costanzo et al., 2013*; *Pearce et al., 2015*; *Pecho-Vrieseling et al., 2014*; *Ren et al., 2009*). We have previously established transgenic *Drosophila* that employ binary expression systems [e.g., Gal4-UAS, QF-QUAS, or LexA-LexAop (*Riabinina and Potter, 2016*)] to express fluorescent protein (FP) fusions of Htt$_{ex1}$ in non-overlapping cell populations to monitor cell-to-cell transfer of mHtt$_{ex1}$ aggregates in intact brains (*Donnelly and Pearce, 2018*; *Pearce et al., 2015*). Our experimental approach (*Figure 1B*) exploits the previously-reported finding that wtHtt$_{ex1}$ proteins aggregate upon physically encountering mHtt$_{ex1}$ aggregate seeds (*Chen et al., 2001*; *Preisinger et al., 1999*), such that transfer of mHtt$_{ex1}$ aggregates from 'donor' cells is reported by conversion of cytoplasmic wtHtt$_{ex1}$ from its normally soluble, diffuse state to a punctate, aggregated state in 'acceptor' cells (*Figure 1B*-inset). To confirm that mHtt$_{ex1}$ nucleates the aggregation of wtHtt$_{ex1}$ in fly neurons, we co-expressed FP-fusions of these two proteins using pan-neuronal *elav[C155]-Gal4*. In flies expressing only mCherry-tagged mHtt$_{ex1}$ (Htt$_{ex1}$Q91-mCherry) pan-neuronally, aggregates were visible as discrete mCherry+ puncta throughout adult fly brains, with enrichment in neuropil regions (*Figure 1—figure supplement 1A*). By contrast, GFP-tagged wtHtt$_{ex1}$ (Htt$_{ex1}$Q25-GFP) was expressed diffusely in the same regions of age-matched adult brains (*Figure 1—figure supplement 1B*). Upon co-expression with Htt$_{ex1}$Q91-mCherry, Htt$_{ex1}$Q25-GFP was converted to a punctate expression pattern that almost entirely overlapped with Htt$_{ex1}$Q91-mCherry signal (*Figure 1—figure supplement 1C and F*), whereas expression patterns of neither membrane-targeted GFP (mCD8-GFP) nor soluble GFP lacking a polyQ sequence were affected by the presence of Htt$_{ex1}$Q91-mCherry aggregates in neurons (*Figure 1—figure supplement 1D–F*). Thus, in the fly CNS, Htt$_{ex1}$Q91 aggregates induce prion-like conversion of normally-soluble Htt$_{ex1}$Q25 via a homotypic nucleation reaction that requires the Htt$_{ex1}$ sequence.

To examine trans-synaptic prion-like transfer of mHtt$_{ex1}$ aggregates, we coupled QF-driven expression of Htt$_{ex1}$Q91-mCherry with Gal4-driven expression of Htt$_{ex1}$Q25-GFP in neuronal cell populations that make well-defined synaptic connections in the adult fly olfactory system (*Figure 1B–D*). Htt$_{ex1}$Q91-mCherry was expressed using *Or67d-QF* in ~40 presynaptic ORNs ('DA1 ORNs') that project axons from the antenna into the central brain, where they form synaptic connections with dendrites of ~7 partner PNs ('DA1 PNs') in the DA1 glomerulus of the antennal lobe (*Figure 1B-inset, C and D*; *Jefferis et al., 2001*). In these same animals, Htt$_{ex1}$Q25-GFP was expressed in ~60% of PNs using *GH146-Gal4*, which labels lateral and ventral DA1 PNs in addition to other PN types (*Figure 1C and D*; *Marin et al., 2002*). We did not detect expression of Htt$_{ex1}$Q91-mCherry in DA1 ORNs until ~24 hr before eclosion, consistent with activation of adult olfactory receptor gene expression during late pupal development (*Clyne et al., 1999*), whereas Htt$_{ex1}$Q25-GFP was expressed in PNs via *GH146-Gal4* earlier in development (*Stocker et al., 1997*). This genetic approach therefore enables us to monitor prion-like transfer of mHtt aggregates between post-mitotic, synaptically-connected DA1 ORNs and PNs in the adult fly brain.

Formation and prion-like transfer of mHtt$_{ex1}$ aggregates across DA1 ORN-PN synapses was examined by monitoring the solubility of Htt$_{ex1}$Q91-mCherry and Htt$_{ex1}$Q25-GFP proteins in or near the DA1 glomerulus (*Figure 1B-inset and E-G*). Whereas Htt$_{ex1}$Q25-mCherry was expressed diffusely in DA1 ORN axons (*Figure 1C*), aggregated Htt$_{ex1}$Q91-mCherry was visible as discrete puncta almost entirely restricted to DA1 axons and axon termini in the DA1 region of the antennal lobe (*Figure 1D and E1–G1*). We used semi-automated 3D segmentation and reconstruction of high-magnification confocal z-stacks (*Figure 1—figure supplement 2A1* and *Video 1*) to quantify Htt$_{ex1}$Q91 aggregate formation in the DA1 glomerulus over time. Htt$_{ex1}$Q91 aggregates first appeared in pharate adults and increased in number as the flies aged (*Figure 1E1'-G1' and J*). Numbers of Htt$_{ex1}$Q91 aggregates in males exceeded those in females at each time point (*Figure 1J*), consistent with known sexual dimorphism in DA1 glomerular volume (*Stockinger et al., 2005*). In these same brains, Htt$_{ex1}$Q25-GFP was expressed diffusely throughout GH146+ PN cell bodies and processes in young adults (*Figure 1D and E2*), but bright Htt$_{ex1}$Q25-GFP puncta began to appear

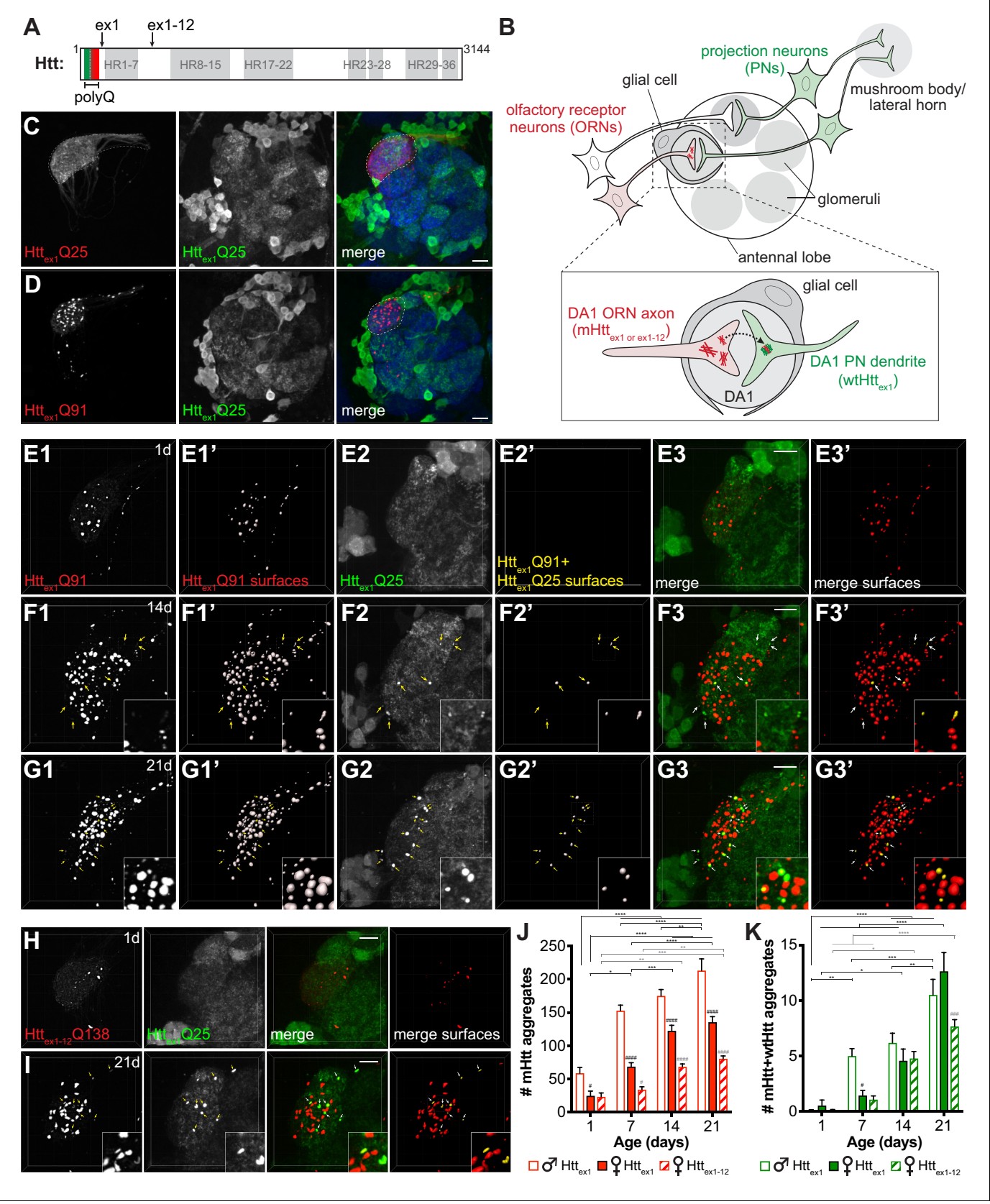

**Figure 1.** mHtt$_{ex1}$ or mHtt$_{ex1-12}$ aggregates formed in presynaptic ORNs induce the aggregation of wtHtt$_{ex1}$ expressed in postsynaptic PNs. (**A**) Primary structure of full-length human Htt (3144 amino acids), including HEAT repeats (HR, *gray regions*) and the N-terminal variable-length polyQ region (*green/red box*), with the pathogenic threshold (~Q37) indicated by a white dotted line. C-termini of two N-terminal mHtt fragments used in this study (Htt$_{ex1}$ and Htt$_{ex1-12}$) are indicated. (**B**) Overall experimental approach. In the fly olfactory system, ORNs synapse with PNs in discrete regions of the antennal lobe known as glomeruli (*gray circles*). PNs send axons into higher brain centers (i.e., mushroom body and/or lateral horn). Draper-expressing glial cells project processes in the antennal lobe, where they ensheath individual glomeruli. To monitor spreading of mHtt aggregates between synaptically-connected ORNs and PNs, we generated transgenic flies that express mHtt$_{ex1}$ or mHtt$_{ex1-12}$ fragments in DA1 ORNs and wtHtt$_{ex1}$ in DA1 PNs. Inset: Transfer of mHtt$_{ex1}$ or mHtt$_{ex1-12}$ aggregates between ORNs and PNs was assessed by monitoring the solubility and colocalization of mHtt and wtHtt fluorescent signals. (**C and D**) Maximum intensity z-projections of antennal lobes from 7 day-old adult males expressing either Htt$_{ex1}$Q25-mCherry (**C**) or Htt$_{ex1}$Q91-mCherry (**D**) in DA1 ORNs using *Or67d-QF* and Htt$_{ex1}$Q25-GFP in GH146+ PNs using *GH146-Gal4*. Raw data are shown in grayscale for individual channels and pseudocolored in merged images. Merged images include Bruchpilot immunofluorescence in blue to mark neuropil, which was used to approximate the boundaries of the DA1 glomerulus (*white dotted lines*). Scale bars = 20 µm. (**E–G**) High-magnification confocal z-stacks of DA1 glomeruli from 1 day-old (**E**), 14 day-old (**F**), and 21 day-old (**G**) adult males expressing Htt$_{ex1}$Q91-mCherry in DA1 ORNs and Htt$_{ex1}$Q25-GFP in GH146+ PNs. Boxed regions in (**F and G**) are shown at higher magnification in insets. Raw data are shown in grayscale in individual channels (Htt$_{ex1}$Q91: **E1, F1, G1**; Htt$_{ex1}$Q25: **E2, F2, G2**) and pseudocolored in merged images (**E3, F3, G3**). mCherry+ 'Htt$_{ex1}$Q91 surfaces' (**E1', F1', G1'**) and 'Htt$_{ex1}$Q91+Htt$_{ex1}$Q25 surfaces' (**E2', F2', G2'**) identified by semi-automated image segmentation are shown adjacent to raw data and pseudocolored *red* and *yellow*, respectively, in the 'merged surfaces' images (**E3', F3', G3'**). Arrows (*yellow* on grayscale images, *white* on merged images) indicate Htt$_{ex1}$Q91+Htt$_{ex1}$Q25 surfaces. Scale bars = 10 µm. (**H and I**) Confocal z-stacks from 1 day-old (**H**) and 21 day-old (**I**) adult females expressing RFP-Htt$_{ex1-12}$Q138 in DA1 ORNs and Htt$_{ex1}$Q25-GFP in GH146+ PNs. Boxed region in (**I**) is shown at higher magnification in insets. RFP+ surfaces identified by semi-automated image segmentation are shown in the last column, with Htt$_{ex1-12}$Q138-only surfaces in *red* and Htt$_{ex1-12}$Q138 +Htt$_{ex1}$Q25 surfaces in *yellow*. Scale bars = 10 µm. (**J and K**) Numbers of Htt$_{ex1}$Q91 or Htt$_{ex1-12}$Q138 ('mHtt') surfaces (**J**) and Htt$_{ex1}$Q91+Htt$_{ex1}$Q25 or Htt$_{ex1-12}$Q138+Htt$_{ex1}$Q25 ('mHtt+wtHtt') surfaces (**K**) identified in adult males (*open bars*) or females (*solid bars*) expressing Htt$_{ex1}$Q91-mCherry in DA1 ORNs or adult females expressing RFP-Htt$_{ex1-12}$Q138 in DA1 ORNs (*striped bars*) at the indicated ages. Data are shown as mean ± SEM; *p<0.05, **p<0.01, ***p<0.001, or ****p<0.0001 by two-way ANOVA followed by Tukey's multiple comparisons tests. '*'s indicate statistical significance comparing flies of the same genotype and sex at different ages (black '*'s compare males or females expressing Htt$_{ex1}$Q91, and gray '*'s compare females expressing Htt$_{ex1-12}$Q138 over time). '#'s indicate statistical significance comparing different genotypes at the same age (black '#'s compare males vs females expressing Htt$_{ex1}$Q91, and gray '#'s compare females expressing Htt$_{ex1}$Q91 vs females expressing Htt$_{ex1-12}$Q138).

The online version of this article includes the following figure supplement(s) for figure 1:

**Figure supplement 1.** mHtt$_{ex1}$ nucleates prion-like conversion of wtHtt$_{ex1}$ in fly neurons.

**Figure supplement 2.** Semi-automatic quantification of seeded wtHtt$_{ex1}$ aggregates.

**Figure supplement 3.** Controls for prion-like transmission of mHtt$_{ex1}$ aggregates from presynaptic DA1 ORNs to postsynaptic PNs.

**Figure supplement 4.** mHtt$_{ex1}$ aggregates do not transfer retrogradely from PN dendrites to ORN axons.

---

and accumulate in the DA1 glomerulus as the flies aged (*Figure 1F2 and G2*, *arrows*). Because these puncta could be difficult to distinguish from surrounding non-aggregated Htt$_{ex1}$Q25-GFP signal, and GFP+ puncta representing normal dendritic architecture and/or intracellular vesicles in the secretory pathway were visible in GH146+ PNs expressing mCD8-GFP (*Figure 1—figure supplement 2B2* and *Figure 1—figure supplement 3B*), we defined Htt$_{ex1}$Q25 aggregates as GFP+ puncta that colocalized with Htt$_{ex1}$Q91 aggregates (*Figure 1—figure supplement 2A1 and C1-7*, and *Video 1*). This approach reported similar results to manual quantification of the same data in 2D confocal slices (*Figure 1—figure supplement 2C1-7 and D*), and identical data were obtained when Htt$_{ex1}$Q25-GFP+ segmented surfaces were filtered for colocalization with Htt$_{ex1}$Q91-mCherry+ puncta (*Figure 1—figure supplement 2A2 and D*). By contrast, Htt$_{ex1}$Q91 aggregates in DA1 ORNs did not colocalize with mCD8-GFP expressed in GH146+ PNs in control animals regardless of whether mCherry+ or GFP+ surfaces were initially segmented (*Figure 1—figure supplement 2B1-2*). Thus, our semi-automated approach to identify 'Htt$_{ex1}$Q91+Htt$_{ex1}$Q25' aggregates specifically reports non-cell autonomous conversion of postsynaptic wtHtt$_{ex1}$ by presynaptic mHtt$_{ex1}$ seeds (*Figure 1B*-inset).

Numbers of Htt$_{ex1}$Q91+Htt$_{ex1}$Q25 aggregates increased as flies expressing Htt$_{ex1}$Q91-mCherry in DA1 ORNs and Htt$_{ex1}$Q25-GFP in GH146+ PNs aged from 1 to 21 days old (*Figure 1E2'-G2' and 1J*). Htt$_{ex1}$Q91 aggregates outnumbered Htt$_{ex1}$Q91+Htt$_{ex1}$Q25 aggregates at each time point tested (*Figure 1J and K*), likely reflecting a higher rate of aggregate formation in 'donor' ORNs expressing polyQ-expanded mHtt$_{ex1}$ than in 'acceptor' PNs, where wtHtt$_{ex1}$ proteins must be nucleated by mHtt$_{ex1}$ seeds originating in other cells. In control experiments, soluble Htt$_{ex1}$Q25-mCherry expressed in DA1 ORNs did not colocalize with Htt$_{ex1}$Q25-GFP expressed in PNs (*Figure 1—figure supplement 3A,E and F*), and Htt$_{ex1}$Q91 aggregates in DA1 ORNs did not nucleate membrane-bound mCD8-GFP in PNs (*Figure 1—figure supplement 3B,E and F*). Htt$_{ex1}$Q91+Htt$_{ex1}$Q25

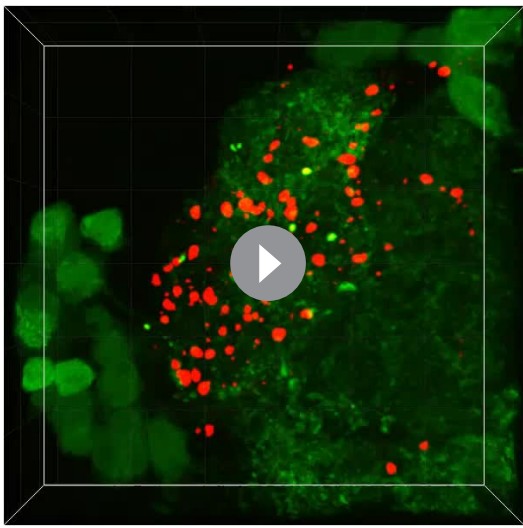

**Video 1.** Semi-automated quantification of mHtt_ex1 and seeded wtHtt_ex1 aggregates in the DA1 glomerulus. Animation illustrating semi-automated approach for quantifying mHtt_ex1 and wtHtt_ex1 aggregates in brains expressing Htt_ex1Q91-mCherry in DA1 ORNs and Htt_ex1Q25-GFP in GH146+ PNs. Data shown in video are the same as in *Figure 1—figure supplement 2A1-2 and C1-7*. Segmentation of raw high-magnification 3D confocal data (0:00) in the red channel (0:07) identified distinct Htt_ex1Q91-mCherry surfaces (0:09), which were filtered for those that co-localize with high-intensity GFP signal to isolate the subpopulation associated with Htt_ex1Q25-GFP puncta (0:13). Volumetric surfaces representing Htt_ex1Q91 (*red*) and Htt_ex1Q91+Htt_ex1Q25 (*yellow*) aggregates (0:18) are shown for each data set analyzed by this method. The animation also illustrates segmentation of raw data in the green channel (0:28) to identify the brightest Htt_ex1Q25-GFP objects in each data set (0:29). Overlap of mCherry+ and GFP+ surfaces, with GFP+ surfaces set at 50% transparency (0:30), highlights co-localization of smaller Htt_ex1Q91 'seeds' surrounded by Htt_ex1Q25 signal in these aggregates.

https://elifesciences.org/articles/58499#video1

aggregates still formed when the Gal4 inhibitor Gal80 was expressed in ORNs (*Figure 1—figure supplement 3C,E and F*) or when the QF repressor QS was expressed in PNs (*Figure 1—figure supplement 3D–F*), arguing strongly against co-expression of Htt_ex1Q91-mCherry and Htt_ex1Q25-GFP using the highly-specific *Or67d-QF* and *GH146-Gal4* drivers. Together, these findings indicate that mHtt_ex1 aggregates formed in pre-synaptic ORNs induce non-cell autonomous, homotypic aggregation of wtHtt_ex1 expressed in the cytoplasm of postsynaptic PNs.

Htt_ex1Q91+Htt_ex1Q25 aggregates were not detected in non-synaptically-connected GH146+ PNs, including in glomeruli directly adjacent to where DA1 ORNs terminate, suggesting that prion-like conversion of Htt_ex1Q25 in PNs by Htt_ex1Q91 aggregates in ORNs requires synaptic connectivity. To examine whether Htt_ex1Q91 aggregates can also spread retrogradely across ORN-PN synapses, we expressed Htt_ex1Q91-mCherry in PNs using *GH146-QF* and Htt_ex1Q25-GFP in all ORNs using *pebbled-Gal4*. While many Htt_ex1Q91 aggregates were visible in PN soma, dendrites, and axons in adult brains (*Figure 1—figure supplement 4A–C*), we did not observe colocalization of Htt_ex1Q25-GFP and Htt_ex1Q91-mCherry puncta within the antennal lobe neuropil in 1, 7, and 14 day-old adults (*Figure 1—figure supplement 4A,B and D*), suggesting that Htt_ex1Q91 aggregate spreading across ORN-PN synapses is restricted to or much more efficient in the anterograde direction. We also found that aggregates formed by mHtt_ex1-12 caspase-6 cleavage products (RFP-Htt_ex1-12Q138) (*Figure 1A*; *Graham et al., 2006*), which have previously been shown to spread from ORN axons to more distant (non-PN) neurons in the fly CNS (*Babcock and Ganetzky, 2015*), also transfer anterogradely across DA1 ORN-PN synapses (*Figure 1H and I*). Together, these findings indicate that multiple pathogenic N-terminal mHtt fragments share the ability to spread trans-synaptically in *Drosophila* brains, and the sequences required for prion-like conversion of wtHtt reside within Htt_ex1.

## wtHtt_ex1 aggregates in PNs are seeded by smaller mHtt_ex1 aggregates originating in ORNs

Higher magnification examination of DA1 glomeruli in flies expressing Htt_ex1Q91-mCherry (*Figure 2A and B*) or RFP-Htt_ex1-12Q138 (*Figure 2C and D*) in DA1 ORNs and Htt_ex1Q25-GFP in GH146+ PNs revealed that most (>85%) Htt_ex1Q25-GFP puncta in DA1 PNs colocalized with Htt_ex1Q91-mCherry aggregates. Segmentation in both the red (*Figure 2A1'-D1'*) and green (*Figure 2A2'-D2'*) channels demonstrated that GFP+ surfaces entirely surrounded associated mCherry+ or RFP+ surfaces in these colocalized aggregates (*Figure 2A3'-D3'*). These data suggest that aggregated Htt_ex1Q91 or Htt_ex1-12Q138 proteins form the 'core' of induced Htt_ex1Q25 aggregates and supports our hypothesis that wtHtt_ex1 solubility in PNs is altered upon direct physical

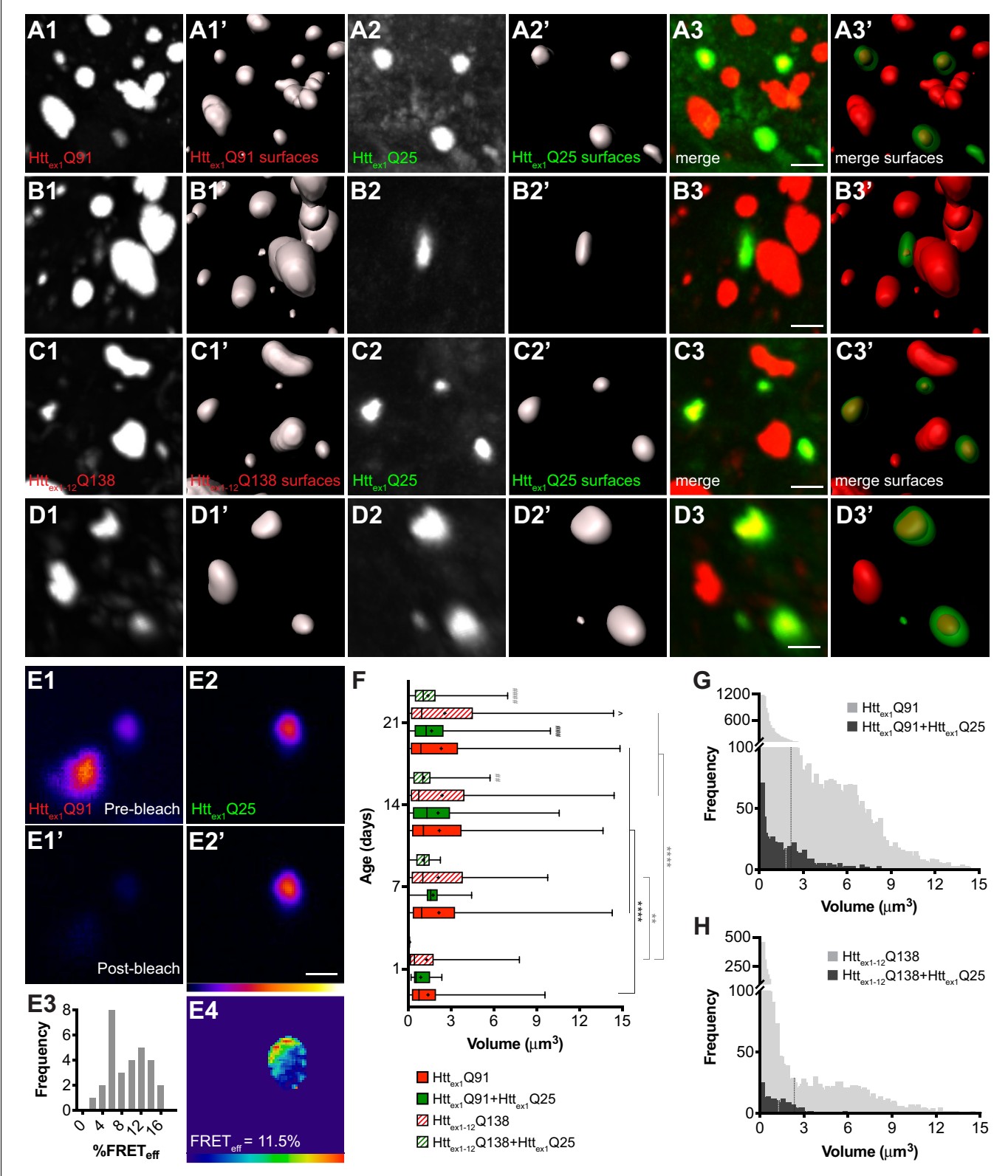

**Figure 2.** wtHtt$_{ex1}$ aggregates in postsynaptic PNs are nucleated by mHtt$_{ex1}$ or mHtt$_{ex1-12}$ aggregates from presynaptic ORNs. (A–D) High-magnification confocal z-stacks of DA1 glomeruli from adult flies expressing Htt$_{ex1}$Q91-mCherry (A and B) or RFP-Htt$_{ex1-12}$Q138 (C and D) in DA1 ORNs and Htt$_{ex1}$Q25-GFP in GH146+ PNs. Raw data (A1-3, B1-3, C1-3, D1-3) are shown adjacent to surfaces identified by 3D segmentation of the red (A1', B1', C1', D1') or green (A2', B2', C2', D2') channels. Htt$_{ex1}$Q25-GFP surfaces are shown at 50% transparency in 'merged surfaces' images (A3', B3', C3',

*Figure 2 continued on next page*

Figure 2 continued

D3') for visibility of co-localized Htt$_{ex1}$Q91-mCherry or RFP-Htt$_{ex1-12}$Q138 surfaces. Scale bars = 1 μm. (E1-4) A single confocal slice through the center of a Htt$_{ex1}$Q91+Htt$_{ex1}$Q25 aggregate before (E1, E2) and after (E1', E2') mCherry acceptor photobleaching. Data are shown as a heat map to highlight changes in fluorescence intensities after photobleaching. Scale bar = 1 μm. FRET efficiency (FRET$_{eff}$) for this aggregate is shown in (E4), and average FRET$_{eff}$ values for all Htt$_{ex1}$Q91+Htt$_{ex1}$Q25 aggregates tested are shown in (E3). (F) Volumes of Htt$_{ex1}$Q91 (*solid red boxes*), Htt$_{ex1}$Q91+Htt$_{ex1}$Q25 (*solid green boxes*), Htt$_{ex1-12}$Q138 (*striped red boxes*), and Htt$_{ex1-12}$Q138+Htt$_{ex1}$Q25 (*striped green boxes*) aggregates identified in the DA1 glomerulus at the indicated ages. Box widths indicate interquartile ranges, vertical lines inside each box indicate medians, whiskers indicate minimums/maximums, and '+'s indicate means for each data set. *p<0.05, **p<0.01, ***p<0.001, ****p<0.0001 by one-way ANOVA followed by Tukey's multiple comparisons test. Statistical significance is indicated by '*'s when comparing the same aggregate sub-population at different ages (Htt$_{ex1}$Q91 surfaces in *black* and Htt$_{ex1-12}$Q138 surfaces in *gray*), by '#'s when comparing Htt$_{ex1}$Q91 vs Htt$_{ex1}$Q91+Htt$_{ex1}$Q25 (*black*) or Htt$_{ex1-12}$Q138 vs Htt$_{ex1-12}$Q138+Htt$_{ex1}$Q25 (*gray*) aggregates at the same ages, and by '˄'s when comparing Htt$_{ex1}$Q91 vs Htt$_{ex1-12}$Q138 aggregates at the same ages. (G and H) Distribution of volumes for (G) Htt$_{ex1}$Q91 (*light gray bars*) and Htt$_{ex1}$Q91+Htt$_{ex1}$Q25 (*dark gray bars*) or (H) Htt$_{ex1-12}$Q138 (*light gray bars*) and Htt$_{ex1-12}$Q138+Htt$_{ex1}$Q25 (*dark gray bars*) aggregates, combined from 7, 14, and 21 day-old flies. Mean volume of Htt$_{ex1}$Q91 or Htt$_{ex1-12}$Q138 aggregates and Htt$_{ex1}$Q91+Htt$_{ex1}$Q25 or Htt$_{ex1-12}$Q138+Htt$_{ex1}$Q25 aggregates are indicated by black and white dotted lines, respectively, on each histogram.

interaction with pre-formed mHtt seeds. To further examine molecular interactions between Htt$_{ex1}$Q91 and Htt$_{ex1}$Q25 proteins, we measured fluorescence resonance energy transfer (FRET) in colocalized aggregates. Remarkably, positive FRET signal was detected for all Htt$_{ex1}$Q91+Htt$_{ex1}$Q25 aggregates analyzed by this method (*Figure 2E*), indicating that the FP tags fused to each of these proteins were in close molecular proximity (<10 nm apart), consistent with direct physical contact.

Comparison of volumes for all segmented Htt$_{ex1}$Q91, Htt$_{ex1-12}$Q138, Htt$_{ex1}$Q91+Htt$_{ex1}$Q25, and Htt$_{ex1-12}$Q138+Htt$_{ex1}$Q25 aggregates revealed that Htt$_{ex1}$Q91 and Htt$_{ex1-12}$Q138 aggregates increased in size as the flies aged, most substantially during the first week of adulthood (*Figure 2F*). Interestingly, the mean volume of Htt$_{ex1}$Q91 or Htt$_{ex1-12}$Q138 surfaces that colocalized with Htt$_{ex1}$Q25 aggregates (herein referred to as 'seeding-competent' mHtt aggregates) was less than the mean volume of all Htt$_{ex1}$Q91 or Htt$_{ex1-12}$Q138 surfaces, and these differences were statistically significant in older animals. When aggregate volumes were analyzed across all time points, it became apparent that seeding-competent Htt$_{ex1}$Q91 or Htt$_{ex1-12}$Q138 aggregates clustered in a subpopulation whose mean was significantly smaller than the mean volume for all Htt$_{ex1}$Q91 or Htt$_{ex1-12}$Q138 aggregates (mean Htt$_{ex1}$Q91+Htt$_{ex1}$Q25 aggregate volume = 1.845 ± 0.068 μm$^3$, mean Htt$_{ex1}$Q91 aggregate volume = 2.19 ± 0.024 μm$^3$, p=0.0005; mean Htt$_{ex1-12}$Q138+Htt$_{ex1}$Q25 aggregate volume = 1.293 ± 0.071 μm$^3$, mean Htt$_{ex1-12}$Q138 aggregate volume = 2.40 ± 0.048 μm$^3$, p<0.0001) (*Figure 2G and H*). We previously reported that Htt$_{ex1}$Q25 aggregates seeded in the cytoplasm of glial cells colocalized with a similarly smaller-sized subpopulation of seeding-competent Htt$_{ex1}$Q91 aggregates from DA1 ORNs (*Pearce et al., 2015*), and smaller mHtt$_{ex1}$ aggregates were associated with increased seeding-propensity and neurotoxicity in other HD models (*Ast et al., 2018*; *Chen et al., 2001*). Taken together, these findings strongly suggest that mHtt$_{ex1}$ or mHtt$_{ex1-12}$ aggregates formed in presynaptic ORNs effect prion-like conversion of wtHtt$_{ex1}$ in the cytoplasm of postsynaptic PNs, and that mHtt aggregate transmissibility in the fly CNS is correlated with smaller aggregate size.

## mHtt$_{ex1}$ aggregate transfer is enhanced across silenced DA1 ORN-PN synapses

Endocytosis, exocytosis, and neuronal activity have been previously implicated in neuron-to-neuron spreading of mHtt and other pathogenic aggregates (*Babcock and Ganetzky, 2015*; *Pecho-Vrieseling et al., 2014*; *Wu et al., 2016*), but it is not known how these processes contribute to aggregate transfer across endogenous synapses in vivo. To examine a role for synaptic activity in DA1 ORN-to-PN transfer of mHtt$_{ex1}$ aggregates, we used well-established fly genetic tools that block fission or fusion of synaptic vesicles at the presynaptic membrane to impair neurotransmission. First, we used *shibire*$^{ts1}$ (*shi*$^{ts1}$) (*Kosaka and Ikeda, 1983*), a temperature-sensitive mutant of the GTPase Shibire/dynamin that blocks endocytic recycling of synaptic vesicles in flies raised at the restrictive temperature. Co-expression of shi$^{ts1}$ with Htt$_{ex1}$Q91-mCherry in DA1 ORNs in flies shifted from the permissive temperature (18°C) to the restrictive temperature (31°C) in adulthood had no effect or slightly decreased numbers of Htt$_{ex1}$Q91 aggregates in the DA1 glomerulus (*Figure 3A–C*), but, surprisingly, strongly enhanced formation of seeded Htt$_{ex1}$Q25 aggregates in DA1 PN dendrites

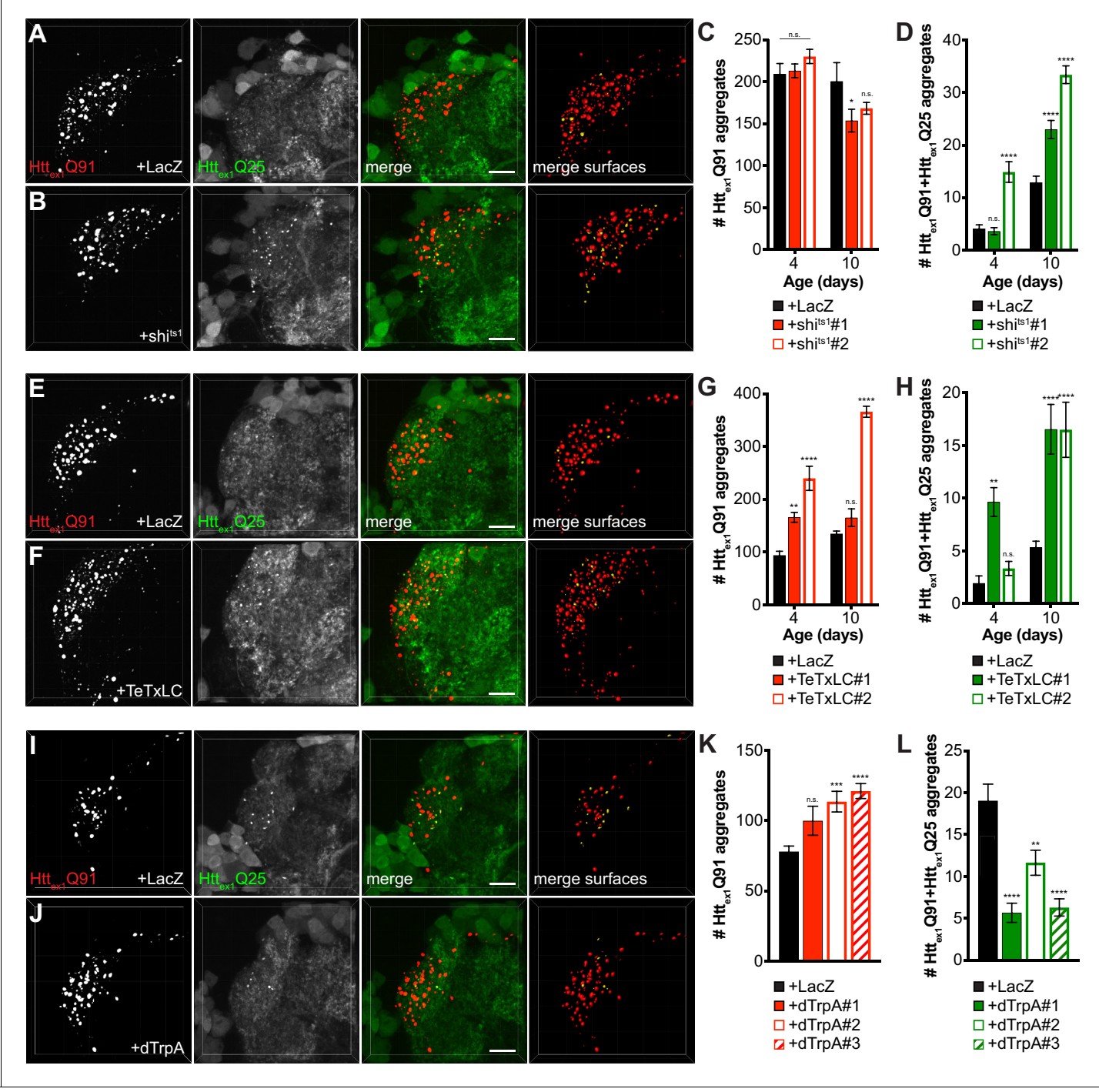

**Figure 3.** mHtt$_{ex1}$ aggregate transfer from ORNs to synaptically-connected PNs is inversely correlated with presynaptic activity. (**A, B, E, F, I, and J**) Confocal z-stacks of DA1 glomeruli from 10 day-old males (**A-B, and E-F**) or 7 day-old females (**I-J**) co-expressing Htt$_{ex1}$Q91-mCherry with either LacZ (**A, E, and I**), shi$^{ts1}$ (**B**), TeTxLC (**F**), or dTrpA (**J**) in DA1 ORNs and Htt$_{ex1}$Q25-GFP in GH146+ PNs. In (**A–B**), flies were raised at the permissive temperature (18°C) and shifted to the restrictive temperature (31°C) upon eclosion, and in (**I–J**), flies were raised at room temperature (~21°C) and shifted to 31°C upon eclosion. mCherry+ surfaces identified by semi-automated image segmentation are shown in the last panels, with Htt$_{ex1}$Q91-only surfaces in *red* and Htt$_{ex1}$Q91+Htt$_{ex1}$Q25 surfaces in *yellow*. Scale bars = 10 µm. (**C-D, G-H, and K-L**) Quantification of Htt$_{ex1}$Q91 (**C, G, and K**) and Htt$_{ex1}$Q91+Htt$_{ex1}$Q25 (**D, H, and L**) aggregates identified in DA1 glomeruli from adult males of the indicated ages co-expressing Htt$_{ex1}$Q91-mCherry with LacZ or shi$^{ts1}$ using two independent *QUAS-shi$^{ts1}$* lines in DA1 ORNs and Htt$_{ex1}$Q25-GFP in GH146+ PNs (**C-D**), adult males of the indicated ages co-expressing Htt$_{ex1}$Q91-mCherry with LacZ or TeTxLC using two independent *QUAS-TeTxLC* lines in DA1 ORNs and Htt$_{ex1}$Q25-GFP in GH146+ PNs (**G-H**), and 7 day-old females expressing Htt$_{ex1}$Q91-mCherry with either LacZ or dTrpA using three independent *QUAS-dTrpA* lines in DA1 ORNs and

*Figure 3 continued on next page*

Figure 3 continued

Htt$_{ex1}$Q25-GFP in GH146+ PNs (**K-L**). Data are shown as mean ± SEM; *p<0.05, **p<0.01, ***p<0.001, ****p<0.0001, 'n.s.' = not significant by one- or two-way ANOVA with Tukey's multiple comparisons test comparing shi$^{ts1}$-, TeTxLC-, or dTrpA-expressing flies to their respective controls expressing LacZ.

compared with control flies expressing LacZ (*Figure 3A,B and D*). Likewise, co-expression of Htt$_{ex1}$Q91-mCherry with tetanus toxin light chain (TeTxLC), which inhibits SNARE-mediated fusion of synaptic vesicles with presynaptic membranes (*Sweeney et al., 1995*), strongly increased numbers of seeded Htt$_{ex1}$Q25 aggregates in the DA1 glomerulus (*Figure 3E,F and H*). At some time points, TeTxLC co-expression led to increased numbers of Htt$_{ex1}$Q91 aggregates compared with control animals (*Figure 3E–G*); however, there appeared to be no correlation between numbers of Htt$_{ex1}$Q91 and Htt$_{ex1}$Q91+Htt$_{ex1}$Q25 aggregates in the DA1 glomerulus over time, so the increased numbers of seeded Htt$_{ex1}$Q25 aggregates were unlikely to be simply due to abundance of presynaptic Htt$_{ex1}$Q91 seeds. To manipulate neuronal activity by an alternative approach, we co-expressed the heat-activated *Drosophila* transient-receptor potential A (dTrpA) channel (*Hamada et al., 2008*) with Htt$_{ex1}$Q91-mCherry to thermogenetically stimulate DA1 ORNs. In adult flies shifted from ~21°C to 31°C upon eclosion, dTrpA-mediated activation of Htt$_{ex1}$Q91-mCherry-expressing DA1 ORNs slightly increased Htt$_{ex1}$Q91 aggregate numbers, but decreased formation of seeded Htt$_{ex1}$Q25 aggregates in the DA1 glomerulus compared with control flies expressing LacZ (*Figure 3I–L*). Together, these results indicate that prion-like transmission of Htt$_{ex1}$Q91 aggregates from ORNs to PNs is inversely correlated with presynaptic ORN activity. These findings suggest that aggregate transfer could be enhanced across dysfunctional synapses, which are an early pathological finding in HD and other neurodegenerative diseases.

Our results using shi$^{ts1}$ and TeTxLC to block DA1 ORN activity contrast with previous reports showing that spreading of Htt$_{ex1-12}$Q138 aggregates was inhibited from endocytosis- or exocytosis-impaired ORNs to non-synaptically-connected neurons (*Babcock and Ganetzky, 2015*) or that botulinum toxin inhibited spreading of mHtt$_{ex1}$ from R6/2 mouse brain slices to functionally-connected human neurons (*Pecho-Vrieseling et al., 2014*). This discrepancy could be due to construct- or cell type-specific effects or possibly different mechanisms regulating synaptic or non-synaptic aggregate transmission in the brain. Thus, we wondered whether blocking Shibire-mediated endocytosis in ORNs might create a more favorable environment for aggregate transfer across endogenous synapses in our HD model. To test this, we first quantified Htt$_{ex1}$Q91-mCherry-expressing DA1 ORN axonal surfaces using mCD8-GFP, a tool widely used to label neuronal cell bodies and processes (*Lee and Luo, 1999*; *Mosca and Luo, 2014*) and to quantify neuron or neurite abundance (*Burr et al., 2014*; *MacDonald et al., 2006*) in fly brains. Segmentation and 3D reconstruction of mCD8-GFP+ DA1 ORN surfaces revealed that fluorescence intensity of and volume occupied by DA1 ORN axons were increased ~2 fold in shi$^{ts1}$-expressing flies at the restrictive temperature compared with controls expressing LacZ (*Figure 4A–C*). This effect appeared to be specific since Htt$_{ex1}$Q91 aggregate abundance in DA1 ORNs was not increased by shi$^{ts}$ co-expression (*Figure 3C*), and may reflect a homeostatic compensatory response to endocytic blockade (*Davis, 2013*; *Dickman et al., 2006*) that could create additional exit sites for Htt$_{ex1}$Q91 aggregates.

To test whether inhibition of Shibire-mediated endocytosis in ORNs has similar effects on ORN-to-glia transfer of Htt$_{ex1}$Q91 aggregates (*Pearce et al., 2015*), we co-expressed shi$^{ts1}$ with Htt$_{ex1}$Q91-mCherry in DA1 ORNs and monitored formation of seeded Htt$_{ex1}$Q25 aggregates in the glial cytoplasm. Similar to effects of shi$^{ts1}$ on trans-synaptic Htt$_{ex1}$Q91 aggregate transfer, blocking Shibire-mediated endocytosis in DA1 ORNs increased seeded Htt$_{ex1}$Q25 aggregate formation in glia without affecting Htt$_{ex1}$Q91 aggregate numbers (*Figure 4D–G*). Together, these findings suggest that shi$^{ts}$-mediated silencing of Htt$_{ex1}$Q91-mCherry-expressing DA1 ORNs alters axonal surface area and enhances prion-like transfer of Htt$_{ex1}$Q91 aggregates from presynaptic ORN axons to the cytoplasm of both glia and postsynaptic PNs.

## Glial Draper is required for ORN-to-PN transfer and alters morphology of neuronal mHtt$_{ex1}$ aggregates

The parallel effects of shi$^{ts1}$-mediated endocytic blockade on transfer of Htt$_{ex1}$Q91 aggregates from DA1 ORNs to DA1 PNs and to glia suggest that mHtt$_{ex1}$ aggregate spreading between these

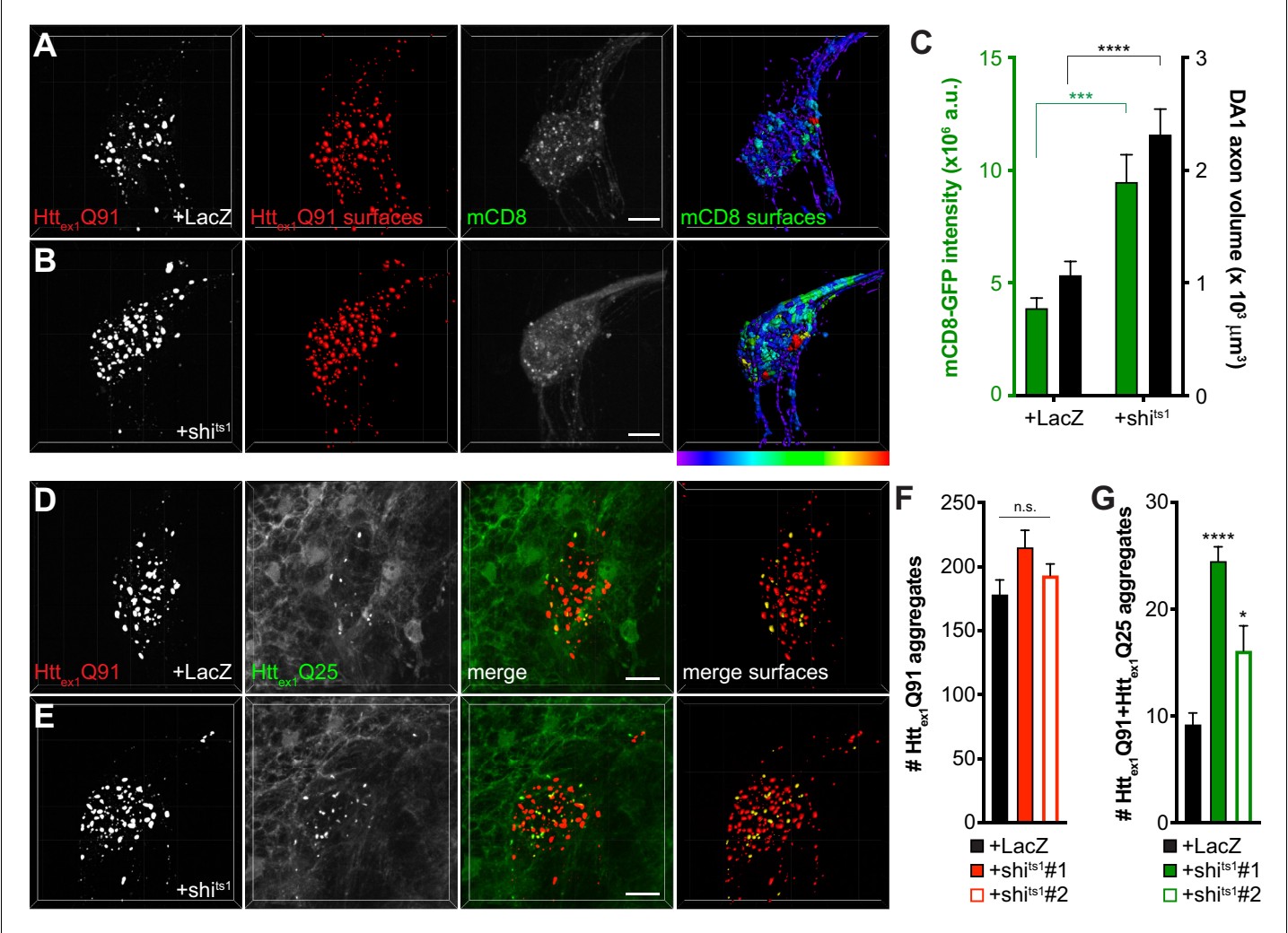

**Figure 4.** Inhibiting Shibire-mediated endocytosis increases mHtt$_{ex1}$-expressing ORN axon volume and enhances transfer of mHtt$_{ex1}$ aggregates from DA1 ORN axons to glia. (**A and B**) Confocal z-stacks of DA1 glomeruli from 10 day-old females co-expressing Htt$_{ex1}$Q91-mCherry, mCD8-GFP, and either LacZ (**A**) or shi$^{ts1}$ (**B**) in DA1 ORNs. Flies were shifted from the permissive temperature (18°C) to the restrictive temperature (31°C) upon eclosion. Raw data are shown in grayscale, and 3D segmented surfaces are shown in *red* for Htt$_{ex1}$Q91 and as a *heat map* for mCD8-GFP to highlight differences in intensity between the genotypes. Scale bars = 10 µm. (**C**) Quantification of mCD8-GFP intensity (left y-axis, *green*) and volume (right y-axis, *black*) of DA1 glomeruli from 10 day-old adult females co-expressing LacZ or shi$^{ts1}$ with Htt$_{ex1}$Q91-mCherry and mCD8-GFP in DA1 ORNs. a.u. = arbitrary units. Data are shown as mean ± SEM; ****p<0.0001 by Student's t-test. (**D and E**) Confocal z-stacks of DA1 glomeruli from 5 to 6 day-old males expressing Htt$_{ex1}$Q91-mCherry with either LacZ (**D**) or shi$^{ts1}$ (**E**) in DA1 ORNs and Htt$_{ex1}$Q25-YFP in repo+ glia. Adult flies were shifted from 18°C to 31°C upon eclosion. mCherry+ surfaces identified by semi-automated image segmentation are shown in the last panels, with Htt$_{ex1}$Q91-only surfaces in *red* and Htt$_{ex1}$Q91+Htt$_{ex1}$Q25 surfaces in *yellow*. Scale bars = 10 µm. (**F and G**) Quantification of Htt$_{ex1}$Q91 (**F**) and Htt$_{ex1}$Q91+Htt$_{ex1}$Q25 (**G**) aggregates in DA1 glomeruli of 5–6 day-old males expressing LacZ or shi$^{ts1}$ using two independent *QUAS-shi$^{ts1}$* lines. Data are shown as mean ± SEM; *p<0.05, ****p<0.0001 by one-way ANOVA with Tukey's multiple comparisons test comparing shi$^{ts1}$-expressing flies to control flies expressing LacZ.

different cell types is coordinated. Our prior work showed that ORN-to-glia transfer of mHtt$_{ex1}$ aggregates is strictly dependent on Draper (*Pearce et al., 2015*), a scavenger receptor responsible for phagocytic engulfment and clearance of neuronal debris in the fly CNS and other tissues (*Etchegaray et al., 2016.*; *Han et al., 2014*; *Hoopfer et al., 2006*; *MacDonald et al., 2006*). Therefore, we sought to determine whether Draper-expressing phagocytic glia might play a role in transferring mHtt$_{ex1}$ aggregates from ORNs to PNs. To test this, we quantified numbers of Htt$_{ex1}$Q91 and seeded Htt$_{ex1}$Q25 aggregates in DA1 ORN axons and PN dendrites, respectively, in animals heterozygous or homozygous for the *draper(drpr)$^{Δ5}$* null mutation (*Freeman et al., 2003*). We previously reported that *drpr* knockout (KO) increased steady-state numbers of Htt$_{ex1}$Q91 aggregates in

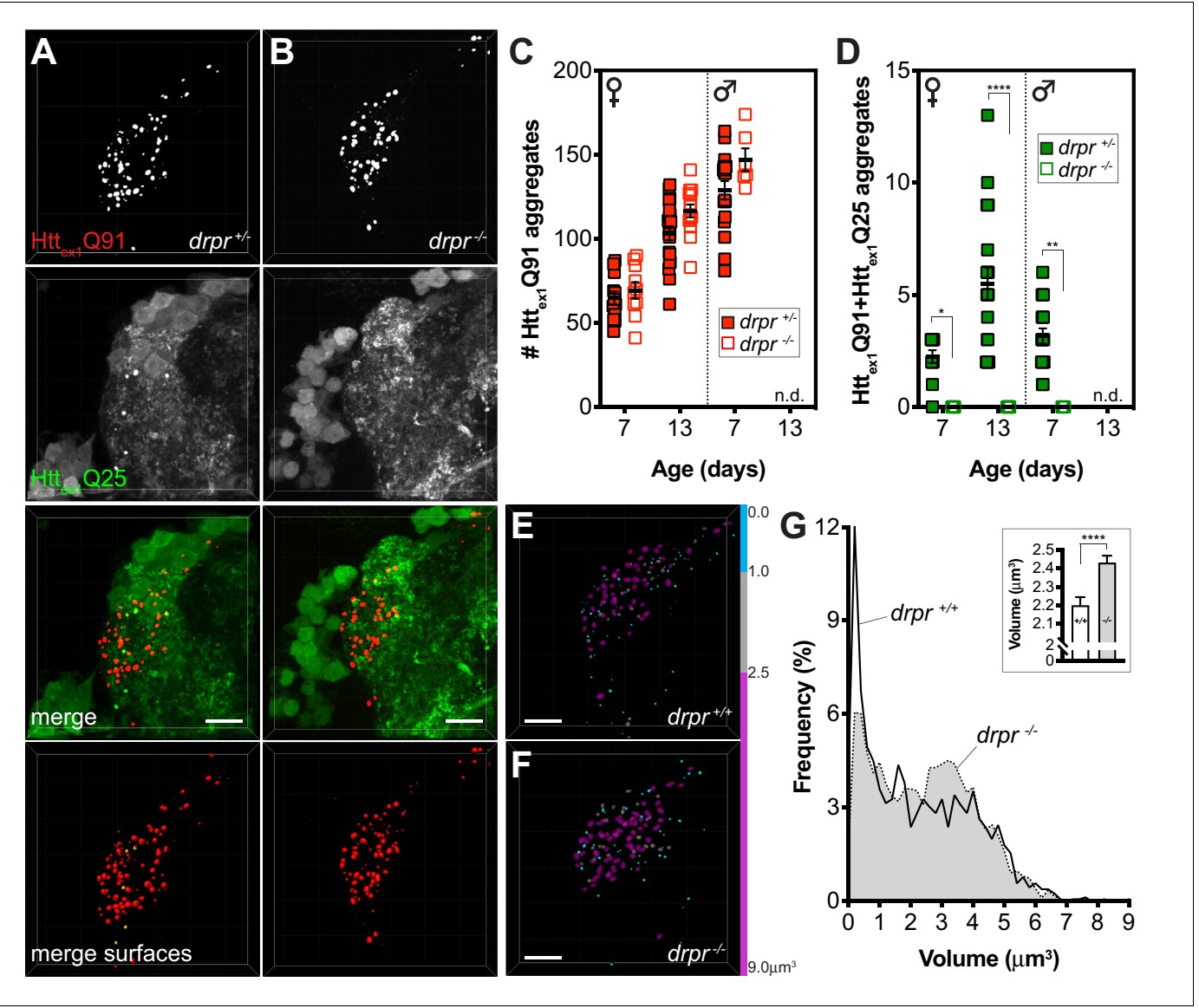

**Figure 5.** Draper mediates mHtt$_{ex1}$ aggregate transfer from presynaptic DA1 ORNs to postsynaptic PNs and regulates neuronal mHtt$_{ex1}$ aggregate size. (A and B) Confocal z-stacks of DA1 glomeruli from 13 day-old adult females expressing Htt$_{ex1}$Q91-mCherry in DA1 ORNs and Htt$_{ex1}$Q25-GFP in GH146+ PNs, either heterozygous (A; *drpr* $^{+/-}$) or homozygous (B; *drpr* $^{-/-}$) for the *drpr*$^{\Delta5}$ null allele. mCherry+ surfaces identified by semi-automated image segmentation are shown in the last row, with Htt$_{ex1}$Q91-only surfaces in *red* and Htt$_{ex1}$Q91+Htt$_{ex1}$Q25 surfaces in *yellow*. Scale bars = 10 μm. (C and D) Quantification of Htt$_{ex1}$Q91 (C) and Htt$_{ex1}$Q91+Htt$_{ex1}$Q25 (D) aggregates in DA1 glomeruli from female or male *drpr* $^{+/-}$ or *drpr* $^{-/-}$ flies at the indicated ages. Data are shown as mean ± SEM; *p<0.05, **p<0.01, ****p<0.0001 by two-way ANOVA with Tukey's multiple comparisons test for *drpr* $^{+/-}$ vs *drpr* $^{-/-}$ flies at the same ages. 'n.d.'=not determined; 13 day-old *drpr* $^{-/-}$ males were not viable. (E and F) Htt$_{ex1}$Q91 surfaces identified in DA1 glomeruli from 7 day-old *drpr* $^{+/-}$ or *drpr* $^{-/-}$ females expressing Htt$_{ex1}$Q91-mCherry in DA1 ORNs. mCherry+ surfaces are color-coded according to the following volume ranges: cyan = 0–1.0 μm³; gray = 1.01–2.49 μm³; magenta = 2.5–9.0 μm³. Gray and magenta surfaces were set to 70% transparency to improve visibility of smaller cyan surfaces. Scale bars = 10 μm. (G) Relative frequency of volumes for all Htt$_{ex1}$Q91 aggregates identified in 7 day-old *drpr* $^{+/+}$ (*solid line*) or *drpr* $^{-/-}$ (*dotted line; gray shading*) males and females. The inset graph shows mean Htt$_{ex1}$Q91 aggregate volume ± SEM for the two genotypes. ****p<0.0001 by unpaired Student's t-test.

The online version of this article includes the following figure supplement(s) for figure 5:

**Figure supplement 1.** Draper is required for enhanced transfer of mHtt$_{ex1}$ from shi$^{ts1}$-expressing DA1 ORNs to GH146+ PNs.

**Figure supplement 2.** Gal80-mediated repression of Gal4 in glia or RNAi knockdown of drpr in PNs do not alter ORN-to-PN prion-like transfer of mHtt$_{ex1}$ aggregates.

**Figure supplement 3.** mHtt$_{ex1}$ aggregates generated in ORNs do not co-localize with markers of lysosomes or autophagosomes in glia.

DA1 ORN axons (*Pearce et al., 2015*); however, this effect was not found to be statistically significant between $drpr^{\Delta 5}$ heterozygotes and homozygotes in this study (*Figure 5A–C*). We suspect this is for two reasons: (a) our image segmentation parameters improved identification of very small aggregates, which we show are more abundant when *drpr* is expressed at normal levels (*Figure 5E–G*), and (b) the $Htt_{ex1}Q91$-mCherry transgene used here expressed at lower levels than the transgene used in our previous study. Thus, $Htt_{ex1}Q91$ aggregates initially form more slowly in DA1 ORN axons (compare numbers of $Htt_{ex1}Q91$ aggregates in young females in *Figure 1J* vs Figure 7I; the latter experiment used the same higher-expressing $Htt_{ex1}Q91$-mCherry transgene as in our prior study) and may be less affected by Draper depletion. Strikingly though, *drpr* KO completely blocked formation of $Htt_{ex1}Q91$+$Htt_{ex1}Q25$ aggregates in postsynaptic PNs (*Figure 5A,B and D*), suggesting that Draper mediates trans-synaptic transfer of $mHtt_{ex1}$ aggregates. This surprising effect of *drpr* KO on seeded $Htt_{ex1}Q25$ aggregate formation was also observed when $shi^{ts1}$ was co-expressed with $Htt_{ex1}Q91$-mCherry in ORNs (*Figure 5—figure supplement 1*), confirming that enhanced transfer of $Htt_{ex1}Q91$ aggregates from $shi^{ts1}$-expressing ORNs occurs via the same Draper-dependent mechanism. To rule out the possibility that *GH146-Gal4* drives expression of $Htt_{ex1}Q25$-GFP in Draper+ glia, we used repo-Gal80 to inhibit Gal4 in all glia and saw no effect on $Htt_{ex1}Q91$ or $Htt_{ex1}Q91$+$Htt_{ex1}Q25$ aggregate numbers (*Figure 5—figure supplement 2A,B,E and F*). Moreover, expression of *drpr*-specific siRNAs in PNs did not affect formation of either $Htt_{ex1}Q91$ or $Htt_{ex1}Q91$+$Htt_{ex1}Q25$ aggregates (*Figure 5—figure supplement 2C–F*), consistent with glia as the sole source of *drpr* expression in the fly CNS. We also did not observe significant colocalization between ORN-derived $Htt_{ex1}Q91$ aggregates and GFP-fusions of Atg8a (*Juhász et al., 2008*) or Lamp1 (*Pulipparacharuvil et al., 2005*) in glia (*Figure 5—figure supplement 3*), suggesting that neuronal $Htt_{ex1}Q91$ aggregates do not seed $Htt_{ex1}Q25$ in glial lysosomes or autophagosomes.

These results point to an unexpected but central role for glial Draper in $Htt_{ex1}Q91$ aggregate transfer between multiple cell types in the fly CNS, but how phagocytic glia could mediate spreading of $Htt_{ex1}Q91$ aggregates across neuronal synapses was not immediately clear. Intriguingly, mean volume of $Htt_{ex1}Q91$ aggregates in DA1 ORN axons was significantly increased in *drpr* KO animals compared to wild-type controls (*Figure 5E,F and G*-inset), and the relative frequency of two aggregate subpopulations shifted between these genotypes: in the absence of *drpr*, the abundance of a smaller-sized subpopulation (~0.1–1 $\mu m^3$) decreased while a larger-sized subpopulation (~2.5–4.0 $\mu m^3$) increased in abundance (*Figure 5E–G*). Intriguingly, the smaller aggregate subpopulation correlated well with the size of $Htt_{ex1}Q91$ aggregates associated with converted $Htt_{ex1}Q25$ in PNs (*Figure 2G*) or glia (*Pearce et al., 2015*), suggesting that phagocytic glia could at least in part mediate formation of smaller seeding-competent $Htt_{ex1}$ aggregates. However, molecular features other than size must regulate the seeding capacity of $mHtt_{ex1}$ aggregates, since smaller-sized aggregates did not completely disappear in *drpr* KO animals (*Figure 5G*). Taken together, these results indicate that Draper-expressing phagocytic glia mediate transfer of $Htt_{ex1}Q91$ aggregates across DA1 ORN-PN synapses, perhaps in part by altering morphological features of neuronal $mHtt_{ex1}$ aggregates.

## Caspase activation in ORNs is required for trans-synaptic transfer of $mHtt_{ex1}$ aggregates

Phagocytic glia engulf injured or degenerating neuronal processes and apoptotic cell corpses by recognizing 'eat me' signals exposed on these debris (*Wilton et al., 2019*), and mHtt expression induces caspase-dependent apoptosis in fly and mammalian models of HD (*Ahmed et al., 2014*). Thus, we asked whether $Htt_{ex1}Q91$-mCherry-expressing DA1 ORNs activate pathways that could stimulate Draper-dependent transfer of aggregates to DA1 PNs. In $drpr^{\Delta 5}$ heterozygotes, expression of $Htt_{ex1}Q91$-GFP in all ORNs did not significantly increase cleavage of *Drosophila* caspase-1 (Dcp-1) compared to flies expressing $Htt_{ex1}Q25$-GFP (*Figure 6A,B and E*); however, Dcp-1 cleavage was significantly increased in $Htt_{ex1}Q91$- vs $Htt_{ex1}Q25$-expressing ORNs when *drpr* was knocked out (*Figure 6C–E*). These data suggest that glia efficiently clear $Htt_{ex1}Q91$-expressing ORN axons displaying Dcp-1-dependent 'eat me' signals via Draper-dependent phagocytosis. In addition, co-expression of the viral effector caspase inhibitor p35 (*Hay et al., 1994*) with $Htt_{ex1}Q91$-mCherry in DA1 ORNs inhibited formation of $Htt_{ex1}Q91$+$Htt_{ex1}Q25$ aggregates in DA1 PNs (*Figure 6F,G,J and K, squares*), indicating that inhibition of apoptotic caspases in presynaptic ORNs phenocopies effects of *drpr* KO on $Htt_{ex1}Q91$ aggregate transfer from ORNs to PNs. By contrast, co-expression of p35 with $Htt_{ex1}Q25$-GFP in PNs did not affect numbers of $Htt_{ex1}Q91$+$Htt_{ex1}Q25$ aggregates in

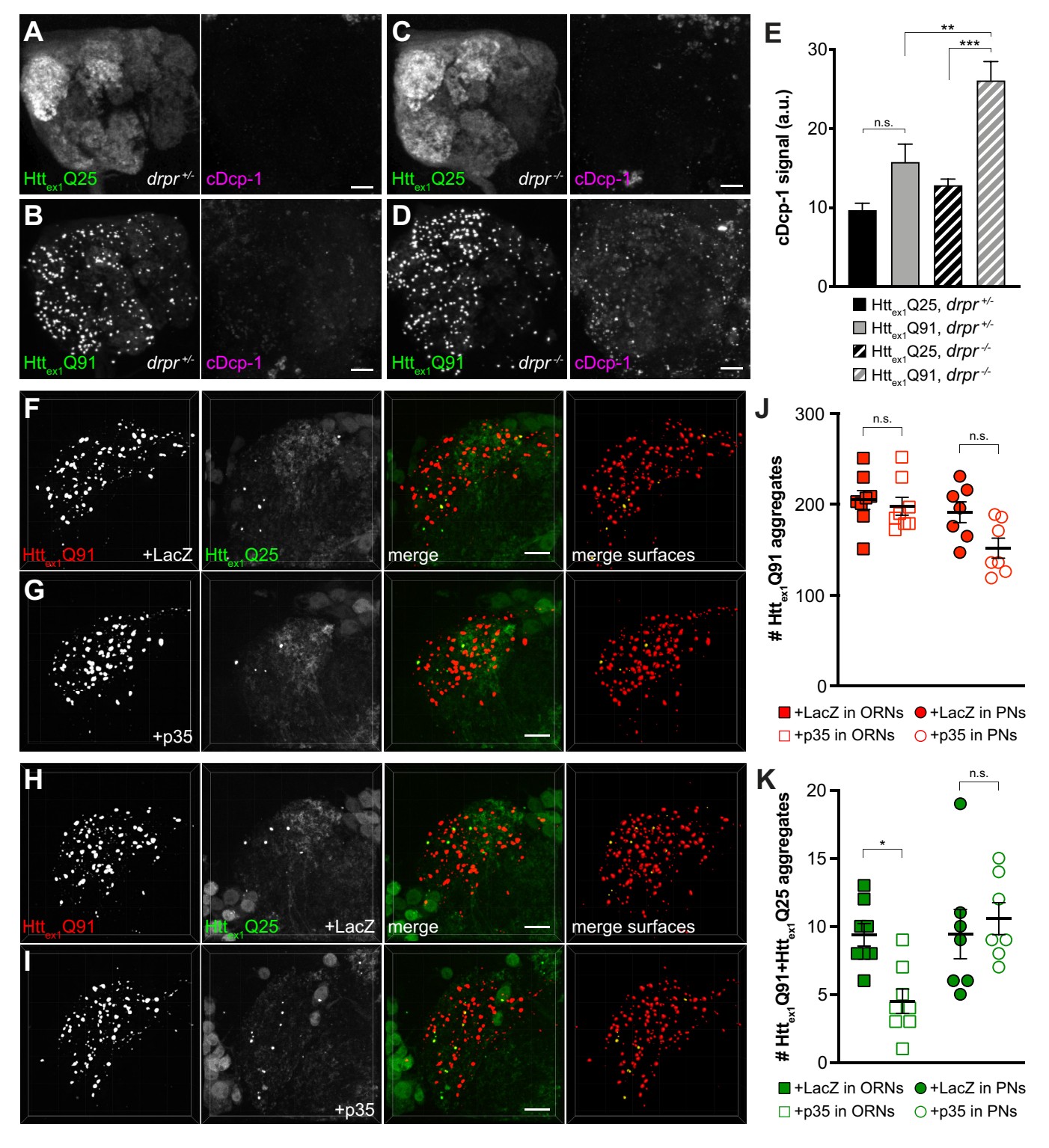

**Figure 6.** Caspase activation in ORNs mediates mHtt$_{ex1}$ aggregate transfer from ORNs to PNs. (A–D) Maximum-intensity projections of antennal lobes from 8 day-old adult males expressing Htt$_{ex1}$Q25-GFP (A and C) or Htt$_{ex1}$Q91-GFP (B and D) in most ORNs using *Or83b-Gal4* in *drpr*$^{\Delta5}$ heterozygotes (*drpr*$^{+/-}$; A and B) or homozygotes (*drpr*$^{-/-}$; C and D). Brains were immunostained for GFP (*left panels*) or cleaved Dcp-1 (*right panels*). Scale bars = 20 µm. (E) Quantification of cDcp-1 immunofluorescence from 8 day-old adult males with the same genotypes as in (A–D). Data are shown as mean ± SEM; **p<0.01, ***p<0.001, 'n.s.' = not significant by one-way ANOVA with Tukey's multiple comparisons test. (F–I) Confocal z-stacks of DA1 glomeruli from 14 day-old males expressing Htt$_{ex1}$Q91-mCherry with LacZ (F) or p35 (G) in DA1 ORNs and Htt$_{ex1}$Q25-GFP in GH146+ PNs, or Htt$_{ex1}$Q91-mCherry in

*Figure 6 continued on next page*

eLife Research article

Cell Biology | Neuroscience

DA1 ORNs and Htt$_{ex1}$Q25-GFP with LacZ (**H**) or p35 (**I**) in GH146+ PNs. mCherry+ surfaces identified by 3D segmentation are shown in the last panels, with Htt$_{ex1}$Q91-only surfaces in *red* and Htt$_{ex1}$Q91+Htt$_{ex1}$Q25 surfaces in *yellow*. Scale bars = 10 μm. (**J and K**) Quantification of Htt$_{ex1}$Q91 (**J**) or Htt$_{ex1}$Q91+Htt$_{ex1}$Q25 (**K**) aggregates in DA1 glomeruli of flies with the same genotypes in (**F and G**) (*squares*) or (**H and I**) (*circles*). Numbers of aggregates in flies expressing LacZ or p35 are indicated by *solid* or *open* shapes, respectively. Data are shown as mean ± SEM; *p<0.05, 'n.s.' = not significant by one-way ANOVA with Tukey's multiple comparisons test.

the DA1 glomerulus (*Figure 6H–K*, *circles*), suggesting that caspase-dependent signaling in PNs is not required for Htt$_{ex1}$Q91 aggregate spreading across ORN-PN synapses. Together, these results suggest that signals mediated by apoptotic caspase activation in DA1 ORNs, and not DA1 PNs, promote engulfment and trans-synaptic transfer of Htt$_{ex1}$Q91 aggregates via phagocytic glia.

## mHtt$_{ex1}$ aggregates transfer from ORNs to PNs via the glial cytoplasm

The strict requirement for glial Draper in prion-like transfer of Htt$_{ex1}$Q91 aggregates across ORN-PN synapses can be explained by two non-mutually exclusive models. In one model (*Figure 7A*, *route 1*), mHtt$_{ex1}$ aggregates spread from presynaptic ORNs to postsynaptic PNs via a glial cytoplasmic intermediate. This model is consistent with our previous finding that phagocytosed neuronal mHtt$_{ex1}$ aggregates gain entry to the glial cytoplasm to effect prion-like conversion of Htt$_{ex1}$Q25 (*Pearce et al., 2015*). Alternatively (*Figure 7A*, *route 2*), phagocytic glia could sculpt the synaptic environment in a way that promotes transfer of mHtt$_{ex1}$ aggregates directly from ORN axons to PN dendrites. To distinguish between these models, we generated transgenic flies that use three binary expression systems (i.e., QF-QUAS, Gal4-UAS, and LexA-LexAop) to independently express a uniquely-tagged Htt$_{ex1}$ transgene in each of three cell populations: Htt$_{ex1}$Q91-mCherry in DA1 ORNs, Htt$_{ex1}$Q25-3xHA in repo+ glia, and Htt$_{ex1}$Q25-YFP in GH146+ PNs (*Figure 7A*). If Htt$_{ex1}$Q91-mCherry aggregates formed in presynaptic ORNs transfer to postsynaptic PNs via the glial cytoplasm, Htt$_{ex1}$Q91-mCherry aggregate seeds should template the aggregation first of Htt$_{ex1}$Q25-3xHA in glia and then Htt$_{ex1}$Q25-YFP in PNs, resulting in appearance of triple-labeled mCherry+/3xHA+/YFP+ puncta in the DA1 glomerulus (*Figure 7A*, *route 1*). A transient double-labeled mCherry+/3xHA+ aggregate subpopulation that have accessed the glial cytoplasm but not yet reached PNs might also be observed in this scenario. Alternatively, if Htt$_{ex1}$Q91-mCherry aggregates do not access the glial cytoplasm *en route* to PNs, only double-labeled mCherry+/3xHA+ and mCherry+/YFP+ aggregates would appear in the DA1 glomerulus (*Figure 7A*, *route 2*).

Brains from transgenic flies expressing these differentially-tagged Htt$_{ex1}$ transgenes in ORNs, glia, and PNs were analyzed, and expression patterns consistent with known morphologies of these cell types in the antennal lobe were observed by confocal microscopy (*Figure 7B and C*, and *Figure 7—figure supplement 1*). In contrast to data in other figures, these samples required immunostaining to detect expression of Htt$_{ex1}$Q25-3xHA and Htt$_{ex1}$Q25-YFP expression in glia and PNs, respectively. We found that immunolabeled aggregate subtypes were not amenable to volumetric segmentation, and so we instead manually quantified single-, double-, and triple-labeled Htt$_{ex1}$ aggregates in confocal slices and used line scan intensity profiling to confirm colocalization. Htt$_{ex1}$Q91-mCherry expression in DA1 ORNs was partially diffuse and partially punctate in young (0–1 day-old) adult flies (*Figure 7B,D,E and I*), but became more punctate as the flies aged (*Figure 7C, F,G and I*). mCherry+/3xHA+ puncta representing Htt$_{ex1}$Q91 aggregates that transferred from ORNs to glia first appeared in 2 day-old adults and increased in number over the next ~24 hr (*Figure 7E,F and J*, *blue bars*). Remarkably, mCherry+/3xHA+/YFP+ puncta also began to appear in 2 day-old adults and increased in abundance until the flies were ~4 days old (*Figure 7F,G and J*, *striped bars*). The timing of appearance of these different aggregate subtypes supports our hypothesis that Htt$_{ex1}$Q91 seeds originating in DA1 ORN axons transit through the glial cytoplasm before reaching the cytoplasm of DA1 PN dendrites (*Figure 7A*, *route 1*). mCherry+/YFP+ puncta representing aggregates that had transferred from DA1 ORNs to DA1 PNs without accessing the glial cytoplasm were not detected in these brains, arguing against direct transfer of Htt$_{ex1}$Q91-mCherry aggregates from ORNs to PNs (*Figure 7A*, *route 2*). Control animals expressing Htt$_{ex1}$Q25 proteins in paired combinations of ORNs, glia, or PNs (*Figure 7—figure supplement 1*) confirmed that Htt$_{ex1}$Q25 aggregates in glia or PNs were only detected when Htt$_{ex1}$Q91 was expressed in ORNs. Remarkably, when we used RNAi to specifically knockdown *drpr* in glia, numbers of Htt$_{ex1}$Q91-

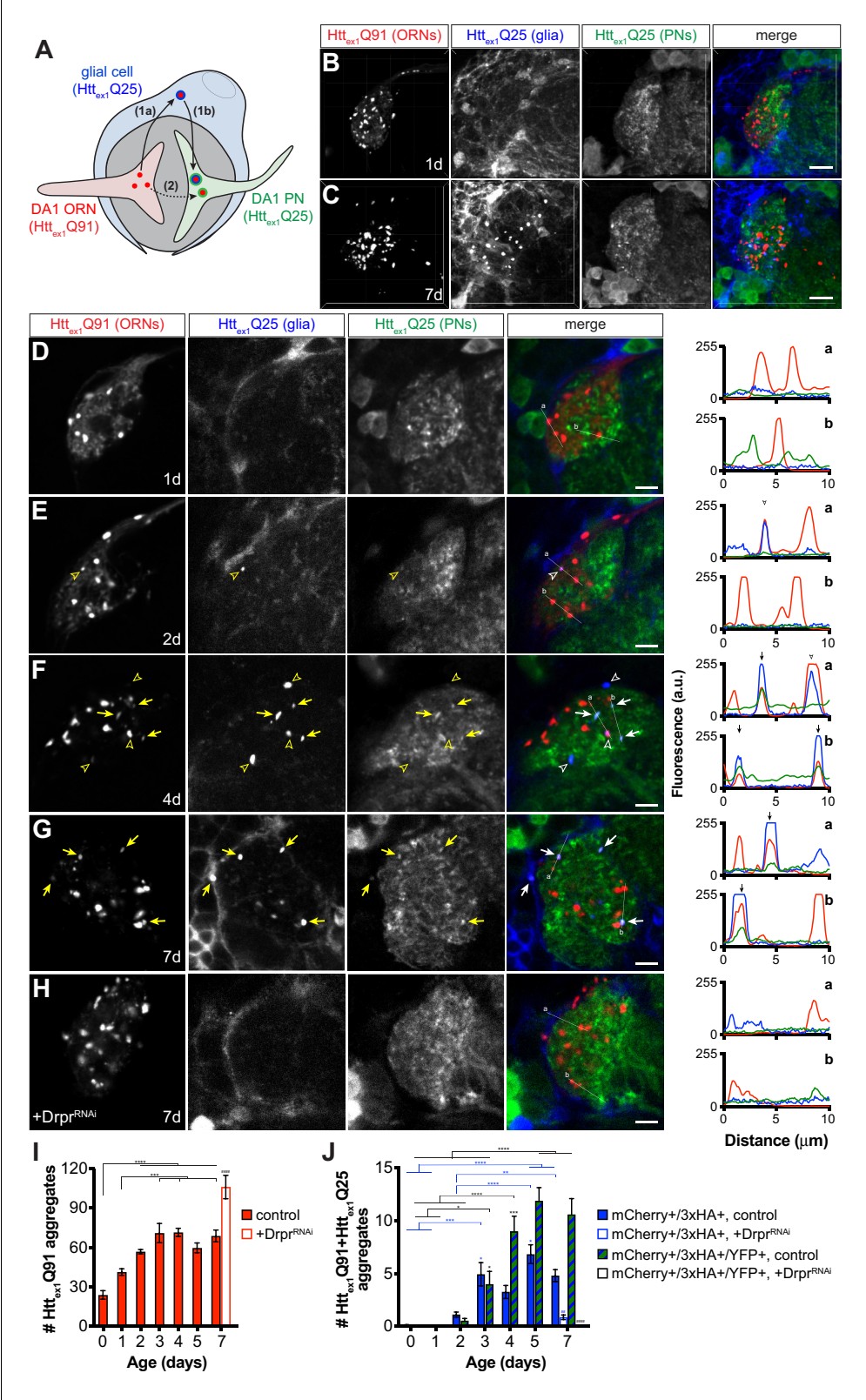

**Figure 7.** mHtt$_{ex1}$ aggregates transfer from presynaptic ORNs to postsynaptic PNs via the cytoplasm of phagocytic glia. (**A**) Diagram illustrating our experimental approach for examining a role for Draper-expressing glia in mHtt$_{ex1}$ aggregate transfer from DA1 ORNs to DA1 PNs. Flies that combined *Or67d-QF-*, *repo-Gal4-*, and *GH146-LexA:GAD*-driven expression of Htt$_{ex1}$Q91-mCherry in DA1 ORNs (*red*), Htt$_{ex1}$Q25-3xHA in all glia (*blue*), and Htt$_{ex1}$Q25-YFP in ~60% of PNs (*green*), respectively, were generated and analyzed by confocal microscopy of immunostained brains. If Htt$_{ex1}$Q91-

*Figure 7 continued on next page*

*Figure 7 continued*

mCherry aggregates travel to PNs via the glial cytoplasm (*route 1*), triple-labeled (mCherry+/3xHA+/YFP+) aggregates should be observed. By contrast, if Htt$_{ex1}$Q91-mCherry aggregates transfer directly to PNs without accessing the glial cytoplasm (*route 2*), then only double-labeled (mCherry+/3xHA+ and mCherry+/YFP+) aggregates would be detected. (B–C) Confocal z-stacks of DA1 glomeruli from adult females of the indicated ages expressing Htt$_{ex1}$Q91-mCherry in DA1 ORNs (*red*), Htt$_{ex1}$Q25-3xHA in glia (*blue*), and Htt$_{ex1}$Q25-YFP in PNs (*green*) at the indicated ages. Brains were immunostained with antibodies against the mCherry, 3xHA, and YFP tags unique to each Htt protein. Scale bars = 10 µm. (D–H) Single 0.35 µm confocal z-slices from females of the indicated ages with (D–G) the same genotype as in (B and C) or (H) also expressing dsRNAs targeting *drpr* in glia ('+Drpr$^{RNAi}$'). Colocalizing mCherry+/3xHA+ or mCherry+/3xHA+/YFP+ aggregates are indicated by open arrowheads or arrows, respectively, shown in yellow on grayscale and white on merged images for increased visibility. Scale bars = 5 µm. Htt$_{ex1}$Q91-mCherry (*red*), Htt$_{ex1}$Q25-3xHA (*blue*), and Htt$_{ex1}$Q25-YFP (*green*) fluorescence intensity profiles for lines 'a' and 'b' are shown to the right of each merged image. Lines were scanned from leftmost to rightmost point. Arrowheads and arrows on graphs indicate peak mCherry fluorescence in colocalized mCherry+/3xHA+ and mCherry+/3xHA+/YFP+ aggregates, respectively. (I and J) Quantification of (I) mCherry-only or (J) mCherry+/3xHA+ and mCherry+/3xHA+/YFP+ aggregates identified in control (*solid bars*) or Drpr$^{RNAi}$-expressing (*open bars*) animals over time. +Drpr$^{RNAi}$ animals were only analyzed at 7 days-old. Data are shown as mean ± SEM; *p<0.05, **p<0.01, ***p<0.001, or ****p<0.0001 by one- or two-way ANOVA followed by Tukey's multiple comparisons test. '*'s indicate statistical significance comparing control flies at different ages [black '*'s compare mCherry-only aggregates in (I) and mCherry+/3xHA+/YFP+ aggregates in (J), and blue '*'s compare mCherry+/3xHA+ aggregates in J], and '#'s indicate statistical significance comparing mCherry+/3xHA+ and mCherry+/3xHA+/YFP+ aggregates, respectively, in control vs Drpr$^{RNAi}$-expressing flies at the same age.

The online version of this article includes the following figure supplement(s) for figure 7:

**Figure supplement 1.** Controls for monitoring transmission of mHtt$_{ex1}$ aggregates from presynaptic DA1 ORNs to postsynaptic PNs via a glial intermediate.

---

mCherry aggregates increased (*Figure 7H and I*), while mCherry+/3xHA+ and mCherry+/3xHA+/YFP+ aggregates significantly decreased or were absent in 7 day-old adults (*Figure 7H and J*). These data confirm that glial Draper mediates Htt$_{ex1}$Q91 aggregate transmission from ORN axons to glia and across ORN-PN synapses. Taken together, these findings suggest that phagocytic glia act as obligatory intermediates in unidirectional prion-like spreading of mHtt$_{ex1}$ aggregates from presynaptic ORNs to postsynaptic PNs in *Drosophila* brains.

## Discussion

The hypothesis that prion-like spreading contributes to progression of protein aggregate pathology in HD and other neurodegenerative diseases is gaining considerable support, and yet we still understand very little about the mechanisms that underlie cell-to-cell spreading in vivo, the influences of cell- and tissue-specific vulnerability, and the relevance of aggregate transmission to disease progression. In this study, we report that mHtt$_{ex1}$ aggregates transfer anterogradely from presynaptic ORNs to postsynaptic PNs via an obligatory path through phagocytic glia in *Drosophila* brains. ORN-to-PN transmission of mHtt$_{ex1}$ aggregates was enhanced by blocking endocytosis and exocytosis and slowed by thermogenetically stimulating presynaptic ORNs, suggesting an inverse relationship between presynaptic activity and mHtt$_{ex1}$ aggregate spreading. mHtt$_{ex1}$ aggregate transfer across synapses was inhibited by blocking apoptosis in presynaptic neurons and required the Draper scavenger receptor, which we have previously reported to mediate phagocytic engulfment of mHtt$_{ex1}$ aggregates from ORN axons and prion-like conversion of wtHtt$_{ex1}$ in the glial cytoplasm (*Pearce et al., 2015*). Here, we expand our understanding of the role that glia play in prion-like diseases by showing that phagocytosed neuronal mHtt$_{ex1}$ aggregates transit through the cytoplasm of glia before reaching postsynaptic PNs. To the best of our knowledge, these findings are the first to uncover a role for a well-conserved phagocytic pathway in prion-like spreading of pathogenic aggregates between neurons in vivo.

The increased propensity of mHtt to aggregate as a result of polyQ expansion is held in check by the proteostasis network, and intrinsic differences in proteostatic capacity of neuronal subpopulations could underlie regional selectivity to inclusion body formation in HD brains (*Margulis and Finkbeiner, 2014*). However, accumulating evidence that mHtt aggregates have prion-like properties suggests that aggregate spreading could also contribute to HD pathogenesis by propagating mHtt pathology through networks of synaptically-connected but selectively-vulnerable neurons. We find that mHtt$_{ex1}$ aggregate transmission across ORN-PN synapses occurs selectively in the anterograde direction and is inversely correlated with synaptic activity. Some of the earliest changes seen in HD patient brains involve loss of presynaptic cortical inputs to the striatum, where more prominent

pathological findings (e.g., mHtt aggregate accumulation and massive loss of medium spiny neurons) appear in later stages of disease (*Reiner and Deng, 2018*). In addition, selective silencing of mHtt in both the cortex and striatum of BACHD mice inhibited striatal degeneration and motor phenotypes to a greater extent that silencing in just the striatum (*Wang et al., 2014*), suggesting that mHtt-induced toxicity in presynaptic cortical regions could play an important role in early HD development. Thus, presynaptic dysfunction may be a driving force for mHtt aggregate spreading between synaptically-connected regions of the brain. While our findings do not directly address secondary consequences of mHtt$_{ex1}$ aggregate spreading across synapses, we identify a novel mechanism whereby glial responses to pathological changes in synapses mediate spreading of toxic aggregates between neurons. How aggregate spreading via glia impacts neuronal viability and functioning at the synaptic and/or circuit level are important questions for future studies.

Glia survey the brain to maintain homeostasis and can rapidly switch between supportive and reactive states in response to perturbations in CNS microenvironments. Upon sensing neuronal insult or injury, reactive microglia and astrocytes undergo dramatic morphological, metabolic, and transcriptional changes and promote neuronal survival by releasing trophic factors and phagocytosing debris (*Hammond et al., 2018*). Central roles for phagocytic glia in neurodegeneration are becoming increasingly recognized as genome-wide association studies and transcriptomic analyses identify glial genes associated with increased disease risk. For example, rare variants in the microglial phagocytic receptor gene *TREM2* are associated with increased risk of AD, FTD, and PD, and loss of TREM2 function exacerbates Aβ-, tau-, and α-synuclein-associated neurotoxicity (*Griciuc et al., 2019*; *Guo et al., 2019*; *Leyns et al., 2019*; *Zhao et al., 2018*). Disruptions in key glial phagocytic functions leads to accumulation of potentially toxic aggregates in the brain (*Asai et al., 2015*; *Hong et al., 2016*; *Pearce et al., 2015*; *Ray et al., 2017*; *Wilton et al., 2019*), and administration of antibodies targeting pathological Aβ, tau, or α-synuclein proteins inhibits aggregate accumulation and spreading in vivo (*Funk et al., 2015*; *Masliah et al., 2005*; *Tran et al., 2014*; *Yanamandra et al., 2013*). However, chronically-active or otherwise dysfunctional glia induce neuroinflammation and exacerbate neuronal damage. For example, hyperactivate microglia excessively engulf synapses in pre-plaque AD mouse brains (*Hong et al., 2016*) and signal for formation of neurotoxic A1 astrocytes, which accumulate in several neurodegenerative diseases (*Liddelow et al., 2017*; *Yun et al., 2018*) and during aging (*Clarke et al., 2018*). Our findings thus add new insights to recognizing glia as double-edged players in neurodegeneration and suggest that rebalancing the protective and harmful effects of glial phagocytosis could be an effective new therapeutic strategy.

We provide several lines of evidence to support a model in which glia engulf mHtt$_{ex1}$ aggregates, or perhaps portions of mHtt$_{ex1}$ aggregate-containing axons, from ORNs without internalizing elements of PNs: (i) mHtt$_{ex1}$ aggregate transmission occurred exclusively in the anterograde direction across ORN-PN synapses, (ii) neuronal mHtt$_{ex1}$ aggregates did not colocalize with markers of lysosomes or autophagosomes in glia, and (iii) caspase inhibition in ORNs and not in PNs inhibited aggregate transfer. Our data suggest that glia selectively target presynaptic ORNs by recognizing apoptotic 'eat me' signals induced by mHtt$_{ex1}$ aggregate accumulation, and this process drives aggregate transfer to postsynaptic PNs. In developing mouse brains, pruned presynaptic structures are engulfed by microglia largely in the absence of postsynaptic markers (*Schafer et al., 2012*; *Weinhard et al., 2018*), suggesting that mammalian glia have the ability to selectively 'nibble' presynaptic components (a fine-tuned phagocytic process known as trogocytosis). In addition, astrocytes internalize dystrophic presynaptic terminals that accumulate near amyloid plaques in transgenic mouse and human AD brains (*Gomez-Arboledas et al., 2018*). Intriguingly, astrocytes expressing the mammalian Draper homolog MEGF10 and complement-dependent microglia preferentially engulf 'weaker' synapses to refine neural circuits in developing and adult mouse brains (*Chung et al., 2013*; *Schafer et al., 2012*), and aberrant activation of these pathways could contribute to early synaptic loss in neurodegenerative disease (*Hong et al., 2016*). We find that cell-to-cell mHtt$_{ex1}$ aggregate transfer is accelerated from silenced ORN axons, suggesting that activity-impaired synapses are engulfed by nearby glia, enhancing spread along the ORN-to-glia-to-PN track. Thus, while the primary objective of phagocytic glia may be to eliminate toxic neuronal debris from the CNS, aberrant and/or excessive activation of phagocytic pathways could paradoxically promote disease. Premature elimination of live synapses could also drive network dysfunction, suggesting that attenuating glial responses could be an effective early intervention to preserve neurological functions in neurodegenerative disease patients (*Carpanini et al., 2019*).

The most exciting yet unexpected finding we report here is that prion-like mHtt$_{ex1}$ aggregates do not directly transfer from ORNs to PNs, but instead make an obligatory detour through the phagocytic glial cytoplasm. This circuitous route could explain why relatively small numbers of seeded wtHtt$_{ex1}$ aggregates form in glia or PNs, since seeding-competent mHtt$_{ex1}$ must escape from multiple degradative systems and from the membrane-bound phagolysosomal compartment in order to access cytoplasmic wtHtt$_{ex1}$. Though transfer through glia is compulsory across ORN-PN synapses, our data cannot exclude the possibility that mHtt$_{ex1}$ aggregates transfer directly between neurons in other regions of the brain or at later stages of disease, when neuron-glia communication and cell integrity could be severely compromised. Our findings suggest that phagocytic glia drive aggregate transfer across ORN-PN synapses by altering the transmissibility of mHtt$_{ex1}$ aggregates formed in presynaptic ORN axons. We previously proposed that Draper-dependent phagocytosis could provide a temporary conduit for engulfed mHtt$_{ex1}$ aggregates to escape into the glial cytoplasm (*Pearce et al., 2015*). This idea is supported by work from others showing that α-synuclein, tau, and mHtt$_{ex1}$ aggregates rupture cell surface or endolysosomal membranes to access the cytoplasmic compartment (*Chen et al., 2019*; *Falcon et al., 2018*; *Flavin et al., 2017*; *Ren et al., 2009*; *Zeineddine et al., 2015*). It is possible that Draper-dependent phagocytosis could modify mHtt$_{ex1}$ aggregates to increase their capacity to cross biological membranes, for example by altering molecular features such as rigidity, frangibility, or size. Indeed, we report here that seeding-competent mHtt$_{ex1}$ aggregates belong to a smaller-sized aggregate subpopulation whose abundance is regulated by Draper, and other groups have observed that the seeding propensity of mHtt$_{ex1}$ (*Ast et al., 2018*; *Chen et al., 2001*) or tau (*Wu et al., 2013*) aggregates is strongly associated with smaller size. This raises the intriguing possibility that membrane fission or fusion events that occur during dynamin-mediated engulfment or phagosome maturation could directly fragment or allow for escape of partially-digested neuronal mHtt$_{ex1}$ aggregates that fall below an upper size limit for entry into the cytoplasm. Indeed, aggregate fragmentation and secondary nucleation events are thought to be key components of prion-like propagation in many neurodegenerative diseases (*Knowles et al., 2014*).

An outstanding question raised by our study is how ORN-derived mHtt$_{ex1}$ aggregates physically transfer from glia to PN cytoplasms. This could be accomplished by a number of mechanisms already proposed for cell-to-cell spreading of prion-like aggregates, such as transport through extracellular vesicles or tunneling nanotubes (*Costanzo et al., 2013*; *Sharma and Subramaniam, 2019*), endocytosis/exocytosis (*Asai et al., 2015*; *Babcock and Ganetzky, 2015*; *Chen et al., 2019*; *Holmes et al., 2013*; *Lee et al., 2010*; *Zeineddine et al., 2015*), secretion or passive release of aggregates from dying cells, and direct penetration of lipid bilayers (*Brundin et al., 2010*; *Davis et al., 2018*; *Vaquer-Alicea and Diamond, 2019*). In a mouse model of tauopathy, pathological tau is transported between anatomically-connected regions of the brain via exosomes secreted by microglia (*Asai et al., 2015*), suggesting that phagocytosed neuronal aggregates may never encounter the extracellular space during transfer. It is also possible that dysfunction caused by continuous aggregate internalization, genetic mutations, and/or normal aging could decrease the efficiency by which glia clear aggregates, promoting their spread. In support of this, extracellular Aβ or α-synuclein fibrils accumulate inside microglia (*Chung et al., 1999*; *Frackowiak et al., 1992*), and the ability of microglia to effectively degrade phagocytosed material declines with age (*Bliederhaeuser et al., 2016*; *Tremblay et al., 2012*). We therefore favor a mechanism whereby phagocytosed aggregates resistant to degradation overwhelm the glial phagolysosomal system and promote aggregate release, either through active secretion (e.g., in an act of self-preservation [*Baron et al., 2017*]) or during glial cell death. While we did not observe mHtt$_{ex1}$ aggregate transmission to non-partner PNs in flies up to 3 weeks old, suggesting that aggregate transfer at these ages is restricted to synaptic regions ensheathed by only 1–2 glial cells (*MacDonald et al., 2006*; *Wu et al., 2017*), important remaining questions are whether aggregates that have invaded the glial cytoplasm can transfer to other cells in or near the DA1 glomerulus (e.g., other glia or local interneurons) or to downstream neurons in the olfactory circuit.

In summary, our data demonstrate that phagocytic glia are active participants in the spread of prion-like protein aggregates between synaptically-connected neurons in vivo. Since microglial processes are highly motile (*Hammond et al., 2018*), these findings raise the intriguing possibility that phagocytic glia could mediate not only trans-synaptic, but also long-range spreading of neuronal aggregate pathology in the mammalian CNS. Our findings have important implications for

understanding complex relationships between aggregate-induced cytotoxicity and neuron-glia communication in health and disease. Deciphering the mechanisms that regulate helpful vs harmful effects of phagocytic glia in the brain will help to reveal the therapeutic potential of targeting key glial functions in HD and other neurodegenerative diseases.

## Materials and methods

### Fly husbandry

All fly stocks and crosses were raised on standard cornmeal/molasses media at 25°C, ~50% relative humidity, and on a 12 hr light/12 hr dark cycle, unless otherwise noted. The following drivers were used to genetically access different cell populations in the fly CNS: *elav[C155]-Gal4* (RRID:BDSC_458; *Lin and Goodman, 1994*), *Or67d-QF* (*Liang et al., 2013*), *GH146-Gal4* (RRID:BDSC_30026; *Stocker et al., 1997*), *pebbled-Gal4* (RRID:BDSC_80570), *GH146-QF* (*Potter et al., 2010*), and *GH146-LexA::GAD* (*Lai et al., 2008*; a kind gift from Tzumin Lee, Janelia Farms), *repo-Gal4* (RRID:BDSC_7415; *Sepp et al., 2001*), and *Or83b-Gal4* (*Kreher et al., 2005*). Transgenic flies previously described by our lab include *QUAS-Htt$_{ex1}$Q91-mCherry*, *QUAS-Htt$_{ex1}$Q25-mCherry*, *UAS-Htt$_{ex1}$Q91-mCherry*, *UAS-Htt$_{ex1}$Q25-GFP*, *UAS-Htt$_{ex1}$Q25-YFP*, and *UAS-GFP* transgenes inserted at the attP3 (1$^{st}$ chromosome; RRID:BDSC_32230) and/or attP24 (2$^{nd}$ chromosome) φC31 integration sites (*Pearce et al., 2015*). Other transgenic flies not generated in this study include: *UAS-mCD8-GFP* (RRID:BDSC_5137), *QUAS-nucLacZ* lines #7 and 44 (RRID:BDSC_30006 and RRID:BDSC_30007), *QUAS-shi$^{ts1}$* lines #2, 5, and 7 (RRID:BDSC_30010, RRID:BDSC_30012; kind gifts from C. Potter, Johns Hopkins School of Medicine), *QUAS-TeTxLC* lines #4c and 9c and *QUAS-dTrpA* lines #5, 6, and 7 (kind gifts from O. Riabinina and C. Potter, Johns Hopkins), *QUAS-mCD8-GFP* line #5J (RRID:BDSC_30002), *QUAS-p35* (a kind gift from H. Steller, Rockefeller University), *UAS-LacZ* (RRID:BDSC_8529), *UAS-p35* (RRID:BDSC_5072), *QUAS-Gal80* (RRID:BDSC_51948), *UAS-QS* (RRID:BDSC_30033), *repo-Gal80* (a kind gift from T. Clandinin, Stanford University), *UAS-FFLuc.VALIUM1* (RRID:BDSC_35789), *UAS-GFP-Lamp1* (RRID:BDSC_42714), *UAS-Atg8a-GFP* (RRID:BDSC_52005), and *UAS-Draper$^{RNAi}$* and *drpr$^{Δ5}$* (RRID:BDSC_67033) flies (kind gifts from M. Freeman, Vollum Institute). Genotypes for all flies used in this study are listed in *Supplementary file 1*.

Cloning and transgenesis pUASTattB(Htt$_{ex1}$Q25-GFP) and pUASTattB(Htt$_{ex1}$Q25-3xHA) plasmids were generated by PCR amplification of tagged Htt$_{ex1}$ cDNAs from the pcDNA3 vector backbone and cloned into pUASTattB via XhoI and XbaI restriction sites. cDNA for mRFP-Htt$_{ex1-12}$Q138 (*Weiss et al., 2012*; a kind gift from T. Littleton, MIT) was subcloned into the pQUASTattB plasmid (*Riabinina et al., 2015* ) using EcoRI and XbaI restriction sites. Htt$_{ex1}$Q25-YFP cDNA was amplified by PCR from the pUASTattB(Htt$_{ex1}$Q25-YFP) plasmid (*Pearce et al., 2015*) and subcloned into the EcoRI site in pLOT downstream of LexA-responsive LexAop DNA sequences. Plasmids were microinjected into embryos containing the attP3 X chromosome (for UAS-Htt$_{ex1}$Q25-GFP), attP3B or attP8 X chromosome (for QUAS-mRFP-Htt$_{ex1-12}$Q138), or attP24 2$^{nd}$ chromosome (for UAS-Htt$_{ex1}$Q25-3xHA or LexAop-Htt$_{ex1}$Q25-YFP) φC31 integration sites either in-house or at BestGene, Inc (Chino Hills, CA).

### *Drosophila* brain dissection and sample preparation

Adult fly brains were dissected, fixed, and stained and/or imaged as previously described (*Pearce et al., 2015*). Briefly, brains were dissected in ice-cold phosphate-buffered saline containing either 0.03% Triton X-100 (PBS/0.03T, when intrinsic fluorescence of FP-fusions was imaged) or 0.3% Triton X-100 (PBS/0.3T, when indirect immunofluorescence was used to detect protein expression). Where possible, we imaged using intrinsic GFP/YFP/mCherry fluorescence when Htt-FP fusions expression levels were high enough to detect on the confocal. Dissected brains were transferred to microfuge tubes containing PBS/T + 4% paraformaldehyde fixative solution on ice and then fixed in the dark at room temperature (RT) for 5 min (when imaging intrinsic fluorescence) or 20 min (when using immunofluorescence). For direct fluorescence imaging, brains were washed in PBS/0.03T buffer several times before incubation in Slowfade Gold Antifade Mountant (Invitrogen, Carlsbad, CA) for at least 1 hr at 4°C in the dark. For immunostaining, brains were washed several times in PBS/0.3T, and then blocked in PBS/0.3T containing 5% normal goat serum (Lampire Biological Laboratories, Pipersville, PA) for 30 min at RT, followed by incubation in primary antibodies diluted in

blocking solution and incubation for 24–72 hr at 4°C in the dark. Brains were then washed in PBS/0.3T several times at RT and incubated in secondary antibodies diluted in blocking solution for 20–24 hr at 4°C in the dark. Following another set of washes in PBS/0.3T at RT, the brains were incubated in Slowfade mountant for at least 16 hr at 4°C in the dark. Brains were then bridge-mounted in Slowfade mountant on glass microscopy slides overlayed with #1.5 coverglass (22 × 22 mm), and edges were sealed using clear nail polish.

Primary antibodies used in this study include rabbit anti-DsRed (RRID:AB_10013483; 1:2000; Takara Bio USA, Inc, Mountain View, CA), rabbit anti-mCherry (RRID:AB_2552323; 1:500; Invitrogen, Carlsbad, CA), chicken anti-GFP (RRID:AB_10000240; 1:500; Aves Labs, Tigard, OR), chicken anti-GFP (RRID:AB_300798; 1:1000; Abcam, Cambridge, UK), chicken anti-GFP (RRID:AB_2534023; 1:500; Invitrogen, Carlsbad, CA), rat anti-HA (clone 3F10; RRID:AB_390918; 1:100; Roche, Basel, Switzerland), rabbit anti-cleaved Dcp-1 (RRID:AB_2721060; 1:100; Cell Signaling Technology, Danvers, MA), and mouse anti-Bruchpilot (clone nc82; RRID:AB_2314866; 1:100; Developmental Studies Hybridoma Bank, Iowa City, IA). Secondary antibodies used include FITC-conjugated donkey anti-chicken (RRID:AB_2340356; 1:200; Jackson Immuno Research Labs, West Grove, PA) and AlexaFluor 488 goat anti-chicken (RRID:AB_2534096), AlexaFluor 568 goat anti-rabbit (RRID:AB_143157), AlexaFluor 647 goat anti-mouse (RRID:AB_2535804), and AlexaFluor 647 goat anti-rat (RRID:AB_141778) IgGs (1:250 each; Invitrogen, Carlsbad, CA).

## Image acquisition

All data were collected on a Leica SP8 laser-scanning confocal system equipped with 405 nm, 488 nm, 561 nm, and 633 nm lasers and 40 × 1.3 NA or 63 × 1.4 NA oil objective lenses. Leica LAS X software was used to establish optimal settings during each microscopy session and to collect optical z-slices of whole-mounted brain samples with Nyquist sampling criteria. Optical zoom was used to further magnify and establish regions of interest in each sample. For most images, confocal data were collected from ~60×60 x 25 μm (*xyz*) stacks centered on a single DA1 glomerulus, which was located using fluorescent signal from Htt$_{ex1}$ protein expressed in DA1 ORN terminals. Exceptions to this are shown in *Figure 1—figure supplement 1A–E* [250 × 250 x~60 μm (*xyz*) stacks capturing fluorescence signal in the anterior central brain] and *Figure 1C–D*, *Figure 1—figure supplement 4A–B*, and *Figure 6A–D* [~150×150 x 30 μm (*xyz*) stacks of the anterior portion of a single antennal lobe].

## Post-imaging analysis

Raw confocal data were analyzed in 2D using ImageJ/FIJI (RRID:SCR_002285; NIH, Bethesda, MD) or in 3D using Imaris (RRID:SCR_007370; Bitplane, Zürich, Switzerland), and all quantitative data were analyzed independently by two researchers blinded to the experimental conditions. For semi-automated quantification of aggregates in DA1 glomeruli, raw confocal data were deconvolved to reduce blur, rendered in 3D, and cropped if necessary to establish the region of interest for further analysis. mHtt fluorescence was segmented in 3D stacks using the 'Surfaces' algorithm (surface detail set to 0.25 μm and background subtraction at 0.75 μm), with the 'split touching objects' option selected, and seed point diameter was set to 0.85 μm. Background thresholding and seed point classification were adjusted manually for each image to optimize segmentation of heterogeneously-sized Htt$_{ex1}$Q91-mCherry puncta ('aggregates') and minimize capturing of diffuse signal. These settings differed <5–10% among individual samples in the same experiment. In rare cases (<5% of aggregates in each image), some larger aggregates were aberrantly split or smaller aggregates in close proximity were incorrectly merged; in these cases, the objects were manually unified or split using the software program. To quantify seeded wtHtt$_{ex1}$ aggregates, red mHtt surfaces that colocalized with Htt$_{ex1}$Q25-GFP were identified by applying a filter for mean intensity in the green channel. Colocalizing aggregates were selected by adjusting the threshold to capture discrete Htt$_{ex1}$Q25 puncta with high contrast compared to surrounding diffuse signal. Segmentation of Htt$_{ex1}$Q25 fluorescence in *Figure 1—figure supplement 2*, *Figure 2A–D*, and *Video 1* was carried out using the same settings described above in the green channel. To measure DA1 ORN axon volume and intensity in *Figure 4A–C*, mCD8-GFP fluorescence was segmented using the 'Surfaces' function in Imaris, with surface detail set to 0.2 μm and background subtraction at 3 μm. Detailed surface measurements (e.g. volume or intensity) were calculated in Imaris, and the data were exported to Excel

(RRID:SCR_016137; Microsoft Corporation, Redmond, WA) or Prism (RRID:SCR_002798; GraphPad Software, San Diego, CA) for further analysis.

For data shown in *Figure 7* and *Figure 7—figure supplement 1*, indirect immunofluorescence was used to detect Htt$_{ex1}$Q25-3xHA expression in glia and amplify low Htt$_{ex1}$Q25-YFP signal in PNs. We found that semi-automated image segmentation as described above reported fewer than half of immunolabeled colocalized aggregates identified by manual counting, likely because of poor anti-body penetration into the aggregate core and Htt$_{ex1}$Q25-YFP aggregates with low contrast that made these data less amenable to 3D segmentation. Instead, we manually counted aggregates in these samples by scanning individual z-slices and scoring discrete puncta with increased signal relative to adjacent diffuse signal in one, two, or all three channels. Line scans confirmed colocalization of signals from the different Htt$_{ex1}$ proteins, as shown in *Figure 7D–H*.

## FRET analysis

Htt$_{ex1}$Q25+Htt$_{ex1}$Q91 colocalized aggregates were analyzed for FRET using the acceptor photo-bleaching method as previously described (*Pearce et al., 2015*). Briefly, 28 individual aggregates were analyzed by photobleaching the mCherry acceptor appended to Htt$_{ex1}$Q91 using a 561 nm laser set at 100% intensity and scanning until fluorescence was no longer detectable. Donor fluorescence dequenching was measured by exciting GFP fused to Htt$_{ex1}$Q25 using a 488 nm laser set at 1% intensity before and after acceptor photobleaching. mCherry fluorescence was also excited before and after photobleaching with a 561 nm laser set at 1% intensity. Fluorescence emission was collected between 500 and 550 nm for GFP and 610 and 700 nm for mCherry to generate before and after images as shown in *Figure 2E*. FRET efficiencies (FRET$_{eff}$) were calculated after back-ground correction using the equation (GFP$_{after}$-GFP$_{before}$)/(GFP$_{after}$) x 100 and represented by a pixel-by-pixel FRET$_{eff}$ image generated using the FRETcalc plugin (*Stepensky, 2007*) in FIJI/ImageJ.

## Statistical analyses

All quantified data were organized and analyzed in Excel or Prism 8. Quantifications in graphical form are shown as mean ± SEM, except for frequency analyses, which are displayed in histograms. Results of all statistical analyses are described in each Figure Legend. Sample size (*n*) for each figure is indicated in *Supplementary file 2* and was selected to yield sufficient statistical power ($\geq$ 5 bio-logical replicates from $\geq$ 3 brains for each condition; a single DA1 glomerulus represents one biolog-ical replicate). Multiple statistical comparisons were performed using the following tests and post-hoc corrections where appropriate: Student's *t*-tests for pairwise comparisons, or one-way or two-way ANOVA followed by Tukey's multiple comparison tests for experiments involving $\geq$3 genotypes. Results of these statistical tests are shown in *Supplementary file 2* and on each graph, and symbols used to indicate statistical significance are defined in the Figure Legends.

## Acknowledgements

The authors would like to thank K Chao, J Hunt, D Luginbuhl, V Reed, B Temsamrit, and M Warkala for technical assistance, G Panning for help with analyzing detailed aggregate measurements, TR Clandinin, MR Freeman, H Krämer, T Lee, JT Littleton, CJ Potter, and O Riabinina for kindly sharing reagents, and members of the Kopito, Luo, and Pearce labs for many valuable discussions. This work was supported by grants from the NIH (R01-DC005982 to LL, R01-NS042842 to RRK, and R03-AG063295 to MMPP), the WW Smith Charitable Trusts (to MMPP), the Pittsburgh Foundation (UN2018-98318 to MMPP), and start-up funds from University of the Sciences (to MMPP).

## Additional information

### Funding

| Funder | Grant reference number | Author |
|---|---|---|
| Pittsburgh Foundation | Integrated Research & Education Grant | Margaret M Panning Pearce |
| W. W. Smith Charitable Trust | Research Grant | Margaret M Panning Pearce |

| National Institutes of Health | R03-AG063295 | Margaret M Panning Pearce |
| National Institutes of Health | R01-DC005982 | Liqun Luo |
| National Institutes of Health | R01-NS042842 | Ron R Kopito |
| University of the Sciences | | Margaret M Panning Pearce |

The funders had no role in study design, data collection and interpretation, or the decision to submit the work for publication.

## Author contributions
Kirby M Donnelly, Formal analysis, Validation, Investigation, Visualization, Methodology, Writing - review and editing; Olivia R DeLorenzo, Aprem DA Zaya, Gabrielle E Pisano, Wint M Thu, Validation, Investigation, Visualization, Writing - review and editing; Liqun Luo, Ron R Kopito, Conceptualization, Resources, Supervision, Funding acquisition, Writing - review and editing; Margaret M Panning Pearce, Conceptualization, Resources, Data curation, Formal analysis, Supervision, Funding acquisition, Validation, Investigation, Visualization, Methodology, Writing - original draft, Project administration, Writing - review and editing

## Author ORCIDs
Liqun Luo (iD) http://orcid.org/0000-0001-5467-9264
Margaret M Panning Pearce (iD) https://orcid.org/0000-0002-5846-9632

## Decision letter and Author response
Decision letter https://doi.org/10.7554/eLife.58499.sa1
Author response https://doi.org/10.7554/eLife.58499.sa2

# Additional files

## Supplementary files
• Supplementary file 1. Full genotypes of flies used in this study.

• Supplementary file 2. Sample sizes and statistical analyses used in this study. Symbols and colors in 'Significance' column match those shown in figures.

• Transparent reporting form

## Data availability
All data generated or analyzed during this study are included in the manuscript and supporting files.

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
