## [Decision Letter]

**Acceptance summary:**

The manuscript provides interesting evidence for glial transfer of pathogenic Htt to seed non-pathogenic Htt aggregation as a prion-like mechanism. The authors' main conclusion, that glia and the Draper pathway mediate the prion-like transmission of protein aggregates across neuronal synapses, could help further our understanding of the molecular events responsible for aggregate development, a major pathology of neurodegenerative diseases.

**Decision letter after peer review:**

[Editors’ note: the authors submitted for reconsideration following the decision after peer review. What follows is the decision letter after the first round of review.]

Thank you for choosing to send your work, "Phagocytic glia are obligatory intermediates in transmission of mutant huntingtin aggregates across neuronal synapses", for consideration at *eLife*. Your submission has been assessed by a Senior Editor in consultation with a member of the Board of Reviewing Editors, Hugo Bellen. Although the work is of interest, we regret to inform you that the findings at this stage are too preliminary for further consideration at *eLife*. If you are willing and able to address the comments, we would be willing to reconsider an improved version but with no guarantee that it would be judged acceptable for publication.

A) The major concerns shared by the three reviewers are the quality/validity of your images especially those with the Htt_ex1_Q25-GFP positive puncta in PNs. You need to address these questions by providing a detail description, quantification criteria and probably re-do some of the images and/or quantification.*Reviewer 1:*

1).… clearly far fewer Htt polyQ aggregates are forming in the postsynaptic cell than in the presynaptic cell (where polyQ Htt is expressed). Likewise, I have no idea how to interpret the primary readout of the assays – from 4-12 HttWT-GFP puncta forming in the PNs of interest. How many PN neurons are actually being assayed in a single glomerus? It seems like a very small effect to be honest…

2) Why is there so little overlap between green Q25and red Q91 Htt puncta in many of the figures? I'm surprised that so much aggregation of HttQ25 is occurring with only a tiny bit of the HttQ91 as a seed. What about the somal Q25 aggregates that don't appear to have any Q91? The authors mention some model of splitting, but I don't quite understand how this could occur.*Reviewer 2:*

2) The robustness of the readout, especially on converted wt Htt_ex1_-Q25 aggregates. In this study, the definition on aggregates is primarily based on fluorescent imaging. Compared to the rather easily recognizable mHtt_ex1_-Q91 aggregates, the Htt_ex1_-Q25 aggregates are often less clear and hard to distinguish from background. For example, the Htt_ex1_-Q25 aggregates in Figure 1H, 2D, 5A and 7C (Htt_ex1_-YFP panels). In these images, many uneven GFP signals are present in the background and hard to differentiate with the assumed Htt_ex1_-Q25 aggregates. Were the aggregates defined mainly by the sizes and intensity of the GFP puncta?

As another concern on this issue, it is noticeable that in Figure 4A, the control used in the study, membrane marker mCD8-GFP, also are presented in bright puncta, indicating an uneven membrane distribution of mCD8-GFP marker, although these bright mCD8-GFP puncta should not be defined as aggregates. ON this regard, it should be noted that Htt_ex1_ is also reported to have membrane-association signal (Atwal et al., 2007) and thus its distribution in cells might not be similar as the GFP alone control (Figure 1—figure supplement 1D). Given that the conversion of wt Htt_ex1_-Q25 into aggregated species is a major evidence for the prion model of mHTT aggregate spreading, it is important to have a robust and rigorous definition on aggregates vs non-aggregates.

3) Figure 6A, data on cDCP-1 staining in normal (non-drpr mutant background) Htt_ex1_-Q25 and Q91 brains should be provided.*Reviewer 3:*

1) In some figures, the images shown are not representative of their quantifications (Figure 3G, Figure 4B and Figure 7). Moreover, the size of the puncta in Figure 2D and Figure 5D is "abnormally" big. Under the same magnification, the size of puncta is much smaller in other images.

3) In Figure 1—figure supplement 1 (A, C and D), the pattern of the mCherry staining is not consistent. Do the authors take the images at a similar layer of the z-sections?

4) It is hard to distinguish the Htt_ex1_Q25-GFP positive puncta from the surrounding diffuse signal. In the Materials and methods section, the authors mentioned that they manually counted the puncta. Did they conduct these quantifications in blind of the genotypes?

B) Reviewer 1 raised a concern that the authors do not provide data to indicate that the synaptic transfer of Htt aggregates is actually causing pathology in their model. They should modify their discussion to note that they do not have this evidence.

Reviewer 1:

1) I have no idea how significant this biology actually is for Htt-mediated pathology. There are no functional studies of the PN neurons to determine if they have any defects secondary to this synaptic transfer…

C) All three reviewers suggest the authors to discuss/modify their model about the role of the glia.

Reviewer 1:

3) Given the requirement of synaptic connectivity for aggregate transfer, it is important to think how the glia contribute to this. Are these glia limited specifically to 1 glomerulus, is there no transfer of Htt aggregates between glia? Seems like the glial transit could degrade the synaptic specificity in areas of the brain where you don't have such specificity driven by the olfactory glomeruli structure.

Reviewer 2:

1).… Further, the study did not provide any mechanistic understanding on how glia-to-PN transmission occurs.

Reviewer 3:

2) In Figure 7, the authors do not provide enough evidence to support their argument that Htt_ex1_Q91 aggregates are transferred from ORNs to glia cells (1a) and then to DA1 PNs (1b). There are a few puncta still can be observed in Figure 7C-Htt_ex1_Q25-YFP (7d, +DrprRNAi). This suggests that the expression of the DrprRNAi in glia cells does not completely block the formation of aggregates in the DA1 PNs. Hence, it is possible that a small portion of Htt_ex1_Q91 aggregates can be transferred directly from ORNs to DA1 PNs.

D) Other concerns

Reviewer 2:

1) Limited novelty and mechanistic study. Given the known role of Draper in the ORN-to-Glia aggregate transmission, the new findings on Draper and glia in ORN-to-PN transmission are less unexpected…

4) In Figure 6E, quantification of Htt_ex1_-Q25-GFP puncta, the baseline for "LacZ in PNs" (filled cycle) is already very low, near zero, thus it might not be meaningful to claim that "Conversely, co-expression of p35 with Htt_ex1_Q25-GFP in PNs did not affect numbers of Htt_ex1_Q91-mCherry and Htt_ex1_Q25-GFP aggregates in the DA1 glomerulus (Figure 6D and E, circles), indicating that caspase-dependent signaling in PNs does not regulate mHtt_ex1_ aggregate spreading across ORN-PN synapses." (subsection “Caspase activation in ORNs regulates trans-synaptic transfer of mHtt_ex1_ aggregates”).

These are the major issues that need to be addressed.

Please note that we aim to publish articles with a single round of revision that would typically be accomplished within two months. This means that work that has potential, but in our judgment would need extensive additional work, to be considered further. We do not intend any criticism of the quality of the data or the rigor of the science. We wish you good luck with your work and we hope you will consider *eLife* for future submissions.

Reviewer #1:

This is a nice study demonstrating trans-synaptic transfer of Htt aggregates via a glial intermediate that result in aggregation with wildtype Htt in the postsynaptic cell. The authors provide convincing genetic demonstration of the cell transit pathway and implicate decreased synaptic activity and glial Drapper function in the process. Overall, a very nice demonstration of some interesting biology of Htt's ability to act in a prion-like fashion only between synaptically connected pairs.

A few points of discussion:

1) I have no idea how significant this biology actually is for Htt-mediated pathology. There are no functional studies of the PN neurons to determine if they have any defects secondary to this synaptic transfer – clearly far fewer Htt polyQ aggregates are forming in the postsynaptic cell than in the presynaptic cell (where polyQ Htt is expressed). Likewise, I have no idea how to interpret the primary readout of the assays – from 4-12 HttWT-GFP puncta forming in the PNs of interest. How many PN neurons are actually being assayed in a single glomerus? It seems like a very small effect to be honest. This is my only real concern – the biology is quite clear with the Htt transfer and prion-like effects requiring a glial intermediate. This is very exciting and well documented. The significance to HD pathology or actual PN function is unknown. It could be quite significant or not really relevant for pathology.

2) Why is there so little overlap between green Q25and red Q91 Htt puncta in many of the figures? I'm surprised that so much aggregation of HttQ25 is occurring with only a tiny bit of the HttQ91 as a seed. What about the somal Q25 aggregates that don't appear to have any Q91? The authors mention some model of splitting, but I don't quite understand how this could occur.

3) Given the requirement of synaptic connectivity for aggregate transfer, it is important to think how the glia contribute to this. Are these glia limited specifically to 1 glomerulus, is there no transfer of Htt aggregates between glia? Seems like the glial transit could degrade the synaptic specificity in areas of the brain where you don't have such specificity driven by the olfactory glomeruli structure.

Reviewer #2:

Prion-like transmission of toxic protein aggregates is emerging as a new mechanism in the pathogenesis of neurodegenerative diseases.

Using the well-defined ORP/PN neuronal network in the fly olfactory system, the Pearce group previously showed that Draper, the glial phagocytic receptor, is essential for the transmission of mHTT-derived aggregates from ORN to the surrounding glia, where they can convert the wildtype, normally soluble HTT protein into aggregated species. In this follow up work by Donnelly et al., they further showed a similar Draper-dependent tran-synaptical transmission of the mHTT aggregate from ORN to the connected PN and the conversion of PN-expressed wildtype HTT (wtHtt_ex1_) to aggregated forms, with the glia cytoplasm as the critical aggregate transit center.

Among the other new findings, they showed that:

1) The transmission is preferably in antegrade direction;

2) The smaller mHTT aggregate species are likely the mediator of the transmission;

3) The transmission efficiency is inversely correlated with activities of presynaptic neurons (i.e., ORN), in contrast to reports from a few other earlier studies;

4) The mHTT-induced apoptotic signaling is likely a potentiator/activator of the Draper-mediated aggregate transmission process.

Although some findings are potentially interesting, overall the study is limited with the following concerns:

1) Limited novelty and mechanistic study. Given the known role of Draper in the ORN-to-Glia aggregate transmission, the new findings on Draper and glia in ORN-to-PN transmission are less unexpected. Further, the study did not provide any mechanistic understanding on how glia-to-PN transmission occurs;

2) The robustness of the readout, especially on converteds wt Htt_ex1_-Q25 aggregates. In this study, the definition on aggregates is primarily based on fluorescent imaging. Compared to the rather easily recognizable mHtt_ex1_-Q91 aggregates, the Htt_ex1_-Q25 aggregates are often less clear and hard to distinguish from background. For example, the Htt_ex1_-Q25 aggregates in Figure 1H, Figure 2D, Figure 5A and Figure 7C (Htt_ex1_-YFP panels). In these images, many uneven GFP signals are present in the background and hard to differentiate with the assumed Htt_ex1_-Q25 aggregates. Were the aggregates defined mainly by the sizes and intensity of the GFP puncta?

As another concern on this issue, it is noticeable that in Figure 4A, the control used in the study, membrane marker mCD8-GFP, also are presented in bright puncta, indicating an uneven membrane distribution of mCD8-GFP marker, although these bright mCD8-GFP puncta should not be defined as aggregates. ON this regard, it should be noted that Htt_ex1_ is also reported to have membrane-association signal (Atwal et al., 2007) and thus its distribution in cells might not be similar as the GFP alone control (Figure 1—figure supplement 1D). Given that the conversion of wt Htt_ex1_-Q25 into aggregated species is a major evidence for the prion model of mHTT aggregate spreading, it is important to have a robust and rigorous definition on aggregates vs non-aggregates.

3) Figure 6A, data on cDCP-1 staining in normal (non-drpr mutant background) Htt_ex1_-Q25 and Q91 brains should be provided;

4) In Figure 6E, quantification of Htt_ex1_-Q25-GFP puncta, the baseline for "LacZ in PNs" (filled cycle) is already very low, near zero, thus it might not be meaningful to claim that "Conversely, co-expression of p35 with Htt_ex1_Q25-GFP in PNs did not affect numbers of Htt_ex1_Q91-mCherry and Htt_ex1_Q25-GFP aggregates in the DA1 glomerulus (Figure 6D and E, circles), indicating that caspase-dependent signaling in PNs does not regulate mHtt_ex1_ aggregate spreading across ORN-PN synapses." (subsection “Caspase activation in ORNs regulates trans-synaptic transfer of mHtt_ex1_ aggregates”).

Reviewer #3:

In this manuscript Donnelly et al., use transgenic *Drosophila* that employ binary expression systems (QF-QUAS, Gal4-UAS, or LexA-LexAop) to independently express fluorescent-tagged Htt_ex1_ transgenes (Htt_ex1_Q91 or Htt_ex1_Q25) in three independent cell populations (ORNs, glia or PNs) of the *Drosophila* olfactory system. They found that the aggregation-prone huntingtin (Htt_ex1_Q91) forms aggregates when it is expressed in the ORNs. The Htt_ex1_Q91 aggregates are transferred anterogradely from ORNs to either glia or PNs. This further leads to the aggregation of the wild-type huntingtins (Htt_ex1_Q25) that are expressed in the glia or PNs. These data strongly support the hypothesized model that pathogenic protein aggregates, such as huntingtin; can spread from cells to cells in the brain. Moreover, the authors show that Draper, a *Drosophila* scavenger receptor responsible for recognizing and phagocytosing cellular debris, is required for this process.

In summary, the authors performed clean *Drosophila* genetic assays to support their conclusions. They proved a complete list of genotypes used in their figures in table 1. This greatly helps to explain the complicated genetics behind all of their experiments. Moreover, their observations that the pathogenic huntingtin aggregates are spread from ORNs to glia and PNs of the olfactory system in a prion-like manner is very interesting. Here is a list of a few concerns.

1) In some figures, the images shown are not representative of their quantifications (Figure 3G, Figure 4B and Figure 7). Moreover, the size of the puncta in Figure 2D and Figure 5D is "abnormally" big. Under the same magnification, the size of puncta is much smaller in other images.

2) In Figure 7, the authors do not provide enough evidence to support their argument that Htt_ex1_Q91 aggregates are transferred from ORNs to glia cells (1a) and then to DA1 PNs (1b). There are a few puncta still can be observed in Figure 7C-Htt_ex1_Q25-YFP (7d, +DrprRNAi). This suggests that the expression of the DrprRNAi in glia cells does not completely block the formation of aggregates in the DA1 PNs. Hence, it is possible that a small portion of Htt_ex1_Q91 aggregates can be transferred directly from ORNs to DA1 PNs.

3) In Figure 1—figure supplement 1 (A, C and D), the pattern of the mCherry staining is not consistent. Do the authors take the images at a similar layer of the z-sections?

4) It is hard to distinguish the Htt_ex1_Q25-GFP positive puncta from the surrounding diffuse signal. In the Materials and methods section, the authors mentioned that they manually counted the puncta. Did they conduct these quantifications in blind of the genotypes?

---

## [Author Response]

[Editors’ note: the authors resubmitted a revised version of the paper for consideration. What follows is the authors’ response to the first round of review.]

A) The major concerns shared by the three reviewers are the quality/validity of your images especially those with the Htt_ex1_Q25-GFP positive puncta in PNs. You need to address these questions by providing a detail description, quantification criteria and probably re-do some of the images and/or quantification.Reviewer 1:1).… clearly far fewer Htt polyQ aggregates are forming in the postsynaptic cell than in the presynaptic cell (where polyQ Htt is expressed). Likewise, I have no idea how to interpret the primary readout of the assays – from 4-12 HttWT-GFP puncta forming in the PNs of interest. How many PN neurons are actually being assayed in a single glomerus? It seems like a very small effect to be honest…

The reviewer raises two concerns here. First, why far more Htt_ex1_Q91 aggregates formed in ORN axons than seeded Htt_ex1_Q25 aggregates in PN dendrites. Htt aggregation is self-replicating, and so we expect many more Htt_ex1_Q91 aggregates to form where this highly aggregation-prone protein was expressed. Our findings show that prion-like conversion of Htt_ex1_Q25 in PNs requires Htt_ex1_Q91 aggregates to exit an ORN axon via Draper-dependent phagocytosis and transit through the glial cytoplasm before reaching the PN cytoplasm. On this journey, phagocytosed aggregates must escape several pathways that promote their clearance (e.g. phagolysosomal degradation, autophagy) and must travel across at least one membrane bilayer, so it is not surprising that Htt_ex1_Q25 aggregate formation would be a rare and perhaps unfavorable event. This idea is now added to the Results section and the Discussion section.

Second, the reviewer asks for clarification of the number of postsynaptic PNs assayed in our images. Here we use Or67d-QF to express Htt_ex1_Q91-mCherry in ~40 DA1 ORNs, which synapse onto ~7 DA1 PNs that are among the larger population of PNs labeled by GH146-Gal4. This relevant information has been added to the Results section. Numbers of induced aggregates detected in the DA1 glomerulus over time are thus consistent with 1-2 Htt_ex1_Q25 aggregates forming in each postsynaptic DA1 PN. Importantly, seeded Htt_ex1_Q25 aggregates were only detectable when Htt_ex1_Q91-mCherry (and not Htt_ex1_Q25-mCherry) was expressed in presynaptic ORNs, and this difference is statistically significant (see Figure 1—figure supplement 3A and F).

2) Why is there so little overlap between green Q25and red Q91 Htt puncta in many of the figures? I'm surprised that so much aggregation of HttQ25 is occurring with only a tiny bit of the HttQ91 as a seed. What about the somal Q25 aggregates that don't appear to have any Q91? The authors mention some model of splitting, but I don't quite understand how this could occur.

The reviewer makes a perceptive observation about the ratio of Htt_ex1_Q91mCherry to Htt_ex1_Q25-GFP fluorescence in seeded wtHtt_ex1_ aggregates. Our data indicate that Htt_ex1_Q25 aggregates co-localize with smaller Htt_ex1_Q91 aggregates, giving the appearance of a Htt_ex1_Q91-mCherry “core” inside each Htt_ex1_Q25-GFP punctum. This interesting feature is highlighted in 3D confocal stacks in revised Figure 2A-D and new Video 1, and in confocal slices in new Figure 1—figure supplement 2C (associated text in subsection “wtHtt_ex1_ aggregates in PNs are seeded by smaller mHtt_ex1_ aggregates originating in ORNs” and the Discussion section). This finding suggests that smaller Htt_ex1_Q91 aggregates act as prion-like “seeds,” in agreement with our previous study (Pearce et al., 2015) and with studies from other labs showing that seeding- and internalization-competent Htt_ex1_ (Chen et al., 2001; Ast et al., 2018) and tau (Wu et al., 2013) belong to smaller-sized aggregate subpopulations. Seeded Htt_ex1_Q91+Htt_ex1_Q25 aggregates could grow by continual addition of Htt_ex1_Q25 until all available protein was converted, possibly increasing the ratio of Htt_ex1_Q25:Htt_ex1_Q91 monomers in each aggregate. It is also possible that secondary nucleation events could be at play, where a fragment of a seeded Htt_ex1_Q25 aggregate that has little or no Htt_ex1_Q91 component could propagate. This could explain why a few bright Htt_ex1_Q25-GFP puncta in the DA1 glomerulus do not have associated Htt_ex1_Q91-mCherry signal, though these would represent only a very small fraction (<10-15%) of all seeded wtHtt_ex1_ aggregates in our samples. However, these Htt_ex1_Q25-GFP-only aggregates were excluded from our analyses because we also observed puncta formed by mCD8-GFP in DA1 and non-DA1 antennal lobe glomeruli (see controls in Figure 1—figure supplement 1, Figure 1—figure supplement 2, and Figure 1—figure supplement 3). Because the majority of wtHtt_ex1_ puncta detected in PN soma did not colocalize with mHtt_ex1_, we have removed these data from the revised manuscript.

Reviewer 2:2) The robustness of the readout, especially on converteds wt Htt_ex1_-Q25 aggregates. In this study, the definition on aggregates is primarily based on fluorescent imaging. Compared to the rather easily recognizable mHtt_ex1_-Q91 aggregates, the Htt_ex1_-Q25 aggregates are often less clear and hard to distinguish from background. For example, the Htt_ex1_-Q25 aggregates in Figure 1H, Figure 2D, Figure 5A and Figure 7C (Htt_ex1_-YFP panels). In these images, many uneven GFP signals are present in the background and hard to differentiate with the assumed Htt_ex1_-Q25 aggregates. Were the aggregates defined mainly by the sizes and intensity of the GFP puncta?

The reviewer correctly points out that seeded Htt_ex1_Q25 aggregates can be more difficult to identify than Htt_ex1_Q91 aggregates because of surrounding non-converted (diffuse) Htt_ex1_Q25-GFP fluorescence. To directly address this issue, we have re-analyzed all of our data (except for Figure 7; see below) using a semi-automated quantification approach to first segment Htt_ex1_Q91-mCherry fluorescence in volumetric confocal data into distinct surfaces (“aggregates”), followed by filtering for the subpopulation of these aggregates that is Htt_ex1_Q25-GFP-positive (further detail described in Results section and Materials and methods section). An additional panel showing Htt_ex1_Q91 (red) and “Htt_ex1_Q91+Htt_ex1_Q25” (yellow) surfaces identified for each data set is now included for each applicable figure. A new figure (Figure 1—figure supplement 2) and animation (Video 1) were also added for further detail about this approach. We believe this strategy provides the most objective and rigorous method to quantify prion-like spreading of mHtt in our model.

In the experiments shown in Figure 7, immunostaining to detect Htt_ex1_Q25-3xHA expressed in glia and Htt_ex1_Q25-YFP in PNs precluded use of this semi-automated 3D segmentation approach, since fewer than half of co-localized Htt_ex1_Q91+Htt_ex1_Q25 surfaces identified by manual counting were identified by 3D segmentation. This is likely because of poor antibody penetration and low contrast of immunolabeled aggregates. To reduce bias, two researchers blinded to the genotype and age of the flies manually counted aggregates in these samples, and we used line scan analyses to confirm colocalization of mHtt_ex1_ and wtHtt_ex1_ signals above surrounding signal (intensity profiles shown in Figure 7). Controls showing that wtHtt_ex1_ proteins in glia and PNs do not colocalize with soluble Htt_ex1_Q25-mCherry in DA1 ORNs were added in a new supplemental figure (Figure 7—figure supplement 1), and glial-specific knockdown of Draper eliminated co-localization of any Htt_ex1_ proteins (Figure 7H-J). Further details about image acquisition and post-imaging analyses were added to the Materials and methods section.

As another concern on this issue, it is noticeable that in Figure 4A, the control used in the study, membrane marker mCD8-GFP, also are presented in bright puncta, indicating an uneven membrane distribution of mCD8-GFP marker, although these bright mCD8-GFP puncta should not be defined as aggregates. ON this regard, it should be noted that Htt_ex1_ is also reported to have membrane-association signal (Atwal et al., 2007) and thus its distribution in cells might not be similar as the GFP alone control (Figure S1D). Given that the conversion of wt Htt_ex1_-Q25 into aggregated species is a major evidence for the prion model of mHTT aggregate spreading, it is important to have a robust and rigorous definition on aggregates vs non-aggregates.

The reviewer correctly notes that membrane-targeted mCD8-GFP can appear both diffuse and punctate when expressed in fly neurons, including in DA1 ORNs (Figure 4A and B). We have included new controls in this revised manuscript showing that a few GFP+ puncta appeared when mCD8-GFP was expressed in PNs (Figure 1—figure supplement 2B and Figure 1—figure supplement 3B) or in all neurons (Figure 1—figure supplement 1D). mCD8-GFP puncta also appeared in regions of the brain where Htt_ex1_Q91 was not expressed, thus representing normal neurite architecture and/or mCD8-GFP-labeled intracellular vesicles in the secretory pathway transiting to the plasma membrane. None of these puncta colocalized with Htt_ex1_Q91 aggregates, measured either by Pearson’s correlation coefficient in all neurons (Figure 1—figure supplement 1F) or using our semi-automated method in the DA1 glomerulus (Figure 1—figure supplement 2B and 2D, and Figure 1—figure supplement 3B and 3F). Based on this, we conservatively define seeded Htt_ex1_Q25 aggregates as Htt_ex1_Q91 aggregates that co-localize with punctate Htt_ex1_Q25-GFP. Controls in which Htt_ex1_Q25-mCherry was expressed in DA1 ORNs did not generate any Htt_ex1_Q91+Htt_ex1_Q25 puncta (Figure 1—figure supplement 3A, E, and F, and Figure 7—figure supplement 1A and B). A more detailed description of how we define and quantify Htt_ex1_ aggregates is now included in the Results section (subsections "Prion-like transfer of mHtt aggregates between synaptically-connected neurons in the adult fly olfactory system" and“mHtt_ex1_ aggregates transfer from ORNs to PNs via the glial cytoplasm”), Materials and methods section, and in relevant figure legends.

3) Figure 6A, data on cDCP-1 staining in normal (non-drpr mutant background) Htt_ex1_-Q25 and Q91 brains should be provided.

cDcp-1 immunostaining in *drpr*^D*5*^ heterozygous flies expressing Htt_ex1_Q25-GFP or Htt_ex1_Q91-GFP in is now included as Figure 6A and B (with *drpr* KO data now in Figure 6C and D; associated results in subsection “Caspase activation in ORNs is required for trans-synaptic transfer of mHtt_ex1_ aggregates”). Interestingly, increased cDcp-1 signal in Htt_ex1_Q91- vs Htt_ex1_Q25-expressing ORN axons was statistically significant in *drpr* knockouts but not in drprheterozygous animals. We suspect this is because Htt_ex1_Q91-expressing ORNs display apoptotic “eat me” signals that are efficiently cleared by Draper-dependent phagocytosis in control animals.

Reviewer 3:1) In some figures, the images shown are not representative of their quantifications (Figure 3G, Figure 4B and Figure 7). Moreover, the size of the puncta in Figure 2D and Figure 5D is "abnormally" big. Under the same magnification, the size of puncta is much smaller in other images.

The reviewer raises a concern here about representative images used for four different experiments. First, in the original Figure 3G (now Figure 3I and J) and in several other raw confocal images, seeded Htt_ex1_Q25 aggregates were not easily distinguished from the surrounding diffuse Htt_ex1_Q25-GFP signal in the DA1 and other glomeruli (same issue raised by reviewer 2 above). We now display all raw data for individual channels in grayscale to improve visibility of puncta, and more importantly, have added a panel for each relevant confocal data figure to show 3D surfaces representing segmented Htt_ex1_Q91 and seeded Htt_ex1_Q25 aggregates. Additional text in the Results section and Materials and methods section describe these new details.

Second, original Figure 4B (now Figure 4A and B) showed DA1 ORN axon surfaces defined by mCD8-GFP signal, but did not clearly report differences in fluorescence intensity; these data are now displayed using a heat map LUT to indicate intensity differences between the two genotypes. Third, single z-slices in Figure 7 show aggregate subtypes that appeared in these brains over time; several new slices were selected to better represent the quantitative data, and line scan profiles were added to demonstrate co-localization (or lack thereof) in puncta indicated in each slice. Fourth, volumetric mHtt surfaces shown in original Figure 2D (now Figure 2A1’-3’, 2B1’-3’, 2C1’-3’, and 2D1’-3’) and 5D (now Figure 5E-F) were shown at a different magnification as adjacent raw data; they are now shown at the same magnification to demonstrate that segmented surfaces accurately report aggregate sizes.

3) In Figure 1—figure supplement 1 (A, C and D), the pattern of the mCherry staining is not consistent. Do the authors take the images at a similar layer of the z-sections?

We have addressed reviewer 3’s concern here about the regions of interest displayed in Figure 1—figure supplement 1 by repeating these experiments using consistent image acquisition settings across all samples: confocal stacks were collected starting at a similar depth in each brain (the most anterior section of the antennal lobe), and a 250 x 250 x ~60µm (*xyz*) stack centered on the central brain (capturing the well-defined morphology of antennal lobes and mushroom body axons) was collected for each sample. These details and similar details for all images are now included in subsection “Image Acquisition”.

4) It is hard to distinguish the Htt_ex1_Q25-GFP positive puncta from the surrounding diffuse signal. In the Materials and methods section, the authors mentioned that they manually counted the puncta. Did they conduct these quantifications in blind of the genotypes?

Reviewer 3 raises the same concern as reviewer 2 above regarding objective quantification of seeded Htt_ex1_Q25 aggregates. As described above, we have addressed this issue by (1) employing a rigorous semi-automated method to quantify aggregates by segmentation of raw confocal data in 3D wherever possible, (2) using line scan analyses to objectively report co-localization when manual quantification was used, and (3) averaging counts from 2 researchers blinded to the experimental conditions (i.e., genotypes and ages) who independently quantified each data file. This new information and details of our methods are included in new Figure 1—figure supplement 2, Video 1, and Figure 7 and in the Results section and Materials and methods section.

B) Reviewer 1 raised a concern that the authors do not provide data to indicate that the synaptic transfer of Htt aggregates is actually causing pathology in their model. They should modify their discussion to note that they do not have this evidence.Reviewer 1:1) I have no idea how significant this biology actually is for Htt-mediated pathology. There are no functional studies of the PN neurons to determine if they have any defects secondary to this synaptic transfer…

Reviewer 1 raises an important point here – almost nothing is known about how spreading of mHtt aggregates or for that matter, any other pathogenic protein aggregates, relates to development of pathology in HD or other neurodegenerative diseases between different brain regions. While we agree that this is an essential question, our current study does not make any direct claims about the relevance of our model to HD pathogenesis in humans. Instead, we use mHtt to model aggregate spreading between different CNS cell types in vivo, and our findings make important advances in understanding the basic cell biological mechanisms that underlie prion-like spreading of aggregates in intact brains. Specifically, our findings demonstrate to our knowledge for the first time that pathogenic aggregates spread unidirectionally between synaptically-connected neurons in vivo and unveil an unexpected role for glia in trans-synaptic aggregate transfer. As suggested, we have reworked our Discussion section to better highlight these exciting findings, while also emphasizing that our current study does not directly address functional consequences of mHtt spreading to glia or postsynaptic neurons, an important focus for future studies.

C) All three reviewers suggest the authors to discuss/modify their model about the role of the glia.Reviewer 1:3) Given the requirement of synaptic connectivity for aggregate transfer, it is important to think how the glia contribute to this. Are these glia limited specifically to 1 glomerulus, is there no transfer of Htt aggregates between glia? Seems like the glial transit could degrade the synaptic specificity in areas of the brain where you don't have such specificity driven by the olfactory glomeruli structure.

Reviewer 1 raises several interesting open questions about how glia could direct transfer of aggregates from presynaptic ORNs to postsynaptic PNs. This was the most surprising and exciting finding of our study, and we agree that these questions are fascinating. In our previous study (Pearce et al., 2015), we show that mHtt_ex1_ aggregates formed in DA1 ORN axons are efficiently transferred to the glial cytoplasm via phagocytosis. We show here that glial engulfment of mHtt_ex1_ aggregates is stimulated by caspase-dependent “eat me” signals on presynaptic ORNs, and because retrograde transfer could not be induced by expressing mHtt_ex1_ in PNs, phagocytic glia must interact in unique ways with presynaptic ORN axons to drive trans-synaptic spreading. We have revised our manuscript with a more developed discussion about how glia-synapse interactions could promote this specificity (Discussion section). Whether this specificity is maintained at non-ORN-PN synapses in the brain was not examined in our study, but is an interesting idea for future work. Reported numbers of ensheathing glia that send processes into the antennal lobe suggests that roughly 1-2 of the glia enwrap each individual antennal lobe glomerulus. While we can’t exclude the possibility of glia-to-glia aggregate transfer, this information, together with the restricted localization of seeded wtHtt_ex1_ aggregates in glia or in PNs to the DA1 glomerulus in our model suggests that this is not likely to be a large component of ORN-to-PN aggregate transfer, nor would it change our overall conclusions about the role of glia in trans-synaptic transfer. Further discussion about glial-dependent and -independent modes of aggregate transfer in the CNS were added to the Discussion section.

Reviewer 2:1).… Further, the study did not provide any mechanistic understanding on how glia-to-PN transmission occurs.

The concern raised here by reviewer 2 is about our limited understanding of the mechanistic details underlying the glia-to-PN leg of the journey that mHtt_ex1_ aggregates take across ORN-PN synapses. Similar to the previous concern from reviewer 1, this question is one of the most interesting raised by our study and merits further investigation. Our previous study (Pearce et al., 2015) showed that mHtt_ex1_ aggregates gain access to the glial cytoplasm at some point following Draper-dependent phagocytic engulfment. In our revised manuscript, we have expanded our Discussion section to include possible mechanisms that could specifically support glia-to-PN aggregate transfer, including previously-reported modes of aggregate transfer (e.g., exosomes or aggregate release) and the possibility that this transfer is driven by secondary cytotoxicity induced by aggregate entry into the glial cytoplasm.

Reviewer 3:2) In Figure 7, the authors do not provide enough evidence to support their argument that Htt_ex1_Q91 aggregates are transferred from ORNs to glia cells (1a) and then to DA1 PNs (1b). There are a few puncta still can be observed in Figure 7C-Htt_ex1_Q25-YFP (7d, +DrprRNAi). This suggests that the expression of the DrprRNAi in glia cells does not completely block the formation of aggregates in the DA1 PNs. Hence, it is possible that a small portion of Htt_ex1_Q91 aggregates can be transferred directly from ORNs to DA1 PNs.

Reviewer 3’s concern is regarding our conclusion that glial Draper knockdown blocks formation of Htt_ex1_Q25 aggregates in PNs (original Figure 7C, now Figure 7H). This concern relates directly to another that we addressed earlier about how aggregates are defined and distinguished from surrounding diffuse Htt_ex1_Q25 signal in PN dendrites. In Figure 7, we have included line scan profiles to accompany confocal slice data, to demonstrate that Htt_ex1_Q25-3xHA or Htt_ex1_Q25-YFP signals in glia and PNs, respectively, do not overlap with Htt_ex1_Q91-mCherry puncta in DrprRNAi-expressing brains, and vice versa. Quantified data from multiple DrprRNAi-expressing brains is also now included in the graphs in Figure 7I and J, showing that knockdown of *drpr* in glia increases numbers of Htt_ex1_Q91 aggregates [as we previously reported (Pearce et al., 2015)] but nearly eliminates formation of seeded Htt_ex1_Q25 aggregates in glia and in PNs. These data strongly support our model that ORN-derived Htt_ex1_Q91 aggregates transfer to both glia and PN cytoplasms via Draper-dependent phagocytosis.

D) Other concernsReviewer 2:1) Limited novelty and mechanistic study. Given the known role of Draper in the ORN-to-Glia aggregate transmission, the new findings on Draper and glia in ORN-to-PN transmission are less unexpected…

The reviewer correctly points out that we have previously reported Draper dependent spreading of Htt_ex1_Q91 aggregates from ORNs to glia (Pearce et al., 2015). However, the current manuscript provides three important new findings: (1) Htt_ex1_Q91 aggregates spread anterogradely and not retrogradely across neuronal synapses in vivo, (2) aggregate spreading across ORN-PN synapses is inversely correlated with neuronal activity, and (3) a surprising, central role for phagocytic glia in trans-synaptic aggregate spreading. Since in vivo studies of prion-like spreading have largely involved focal injection of pre-formed aggregates into rodent brains at concentrations that likely far exceed that experienced in the disease state, we feel that a major contribution of our study is using transgenic methods to provide insight into how aggregates spread between individual cell cytoplasms under more physiological conditions (e.g. across naturally-formed synapses). Protein aggregates have been observed in glial cells in human brains and in animal models of many neurodegenerative diseases, but the details for how glia slow and/or enhance disease progression are not understood. We provide to our knowledge the first evidence that a conserved glial phagocytic pathway mediates both clearance and spreading of protein aggregates in intact brains.

4) In Figure 6E, quantification of Htt_ex1_-Q25-GFP puncta, the baseline for "LacZ in PNs" (filled cycle) is already very low, near zero, thus it might not be meaningful to claim that "Conversely, co-expression of p35 with Htt_ex1_Q25-GFP in PNs did not affect numbers of Htt_ex1_Q91-mCherry and Htt_ex1_Q25-GFP aggregates in the DA1 glomerulus (Figure 6D and E, circles), indicating that caspase-dependent signaling in PNs does not regulate mHtt_ex1_ aggregate spreading across ORN-PN synapses." (subsection “Caspase activation in ORNs regulates trans-synaptic transfer of mHtt_ex1_ aggregates”).

Reviewer 2’s concern is regarding the relatively small numbers of Htt_ex1_Q25 aggregates that were present when p35 or LacZ was co-expressed with Htt_ex1_Q25-GFP in PNs (originally graphed in Figure 6E). We suspect this is due to competition of the Htt_ex1_Q25 and LacZ/p35 transgenes for the GH146-Gal4 driver, leading to lower levels of Htt_ex1_Q25 expression. To increase the robustness of these data, we repeated the experiments shown in the original Figure 6C-E in older animals to compare effects when seeded Htt_ex1_Q25 aggregate numbers are increased. These new data are shown in Figure 6F-K and support our previous conclusion that seeding of Htt_ex1_Q25 is affected by p35 expression in ORNs, but not in PNs.